# 2-Rectifications are Enough for Straight Flows: A Theoretical Insight into Wasserstein Convergence

**Saptarshi Roy**   **Vansh Bansal**   **Purnamrita Sarkar**   **Alessandro Rinaldo**
Department of Statistics and Data Sciences, The University of Texas at Austin

## Abstract

Diffusion models have emerged as a powerful tool for image generation and denoising. Typically, generative models learn a trajectory between the starting noise distribution and the target data distribution. Recently (Liu et al., 2023b) proposed *Rectified Flow* (RF), a generative model that aims to learn straight flow trajectories from noise to data using a sequence of convex optimization problems with close ties to optimal transport. If the trajectory is curved, one must use many Euler discretization steps or novel strategies, such as exponential integrators, to achieve a satisfactory generation quality. In contrast, RF has been shown to theoretically straighten the trajectory through successive rectifications, reducing the number of function evaluations (NFEs) while sampling. It has also been shown empirically that RF may improve the straightness in two rectifications if one can solve the underlying optimization problem within a sufficiently small error. In this paper, we make two key theoretical contributions: 1) we provide the first theoretical analysis of the Wasserstein distance between the sampling distribution of RF and the target distribution. Our error rate is characterized by the number of discretization steps and a *new formulation of straightness* stronger than that in the original work. 2) under a mild regularity assumption, we show that for a rectified flow from a Gaussian to any general target distribution with finite first moment (e.g. mixture of Gaussians), two rectifications are sufficient to achieve a straight flow, which is in line with the previous empirical findings. Additionally, we also present empirical results on both simulated and real datasets to validate our theoretical findings. The codes are available at https://github.com/bansal-vansh/rectified-flow.

## 1 Introduction

In recent years, diffusion models have achieved impressive performance across different multi-modal tasks including image (Ho et al., 2022b; Balaji et al., 2022; Rombach et al., 2022), video (Ho et al., 2022a;c; Luo et al., 2023; Wang et al., 2024; Zhou et al., 2022), and audio (Huang et al., 2023; Kong et al., 2020; Liu et al., 2023a; Ruan et al., 2023) generation that leverages the score-based generative model (SGM) framework (Sohl-Dickstein et al., 2015; Ho et al., 2020), which is a key component of large-scale generative models such as DALL-E 2 (Ramesh et al., 2022). The main idea in this framework is to gradually perturb the data according to a pre-defined diffusion process, and then to learn the reverse process for sample generation. Despite its success, the SGM framework incurs significant computational costs because it requires numerous inference steps to generate high-quality samples. The primary reason is that SGM generates sub-optimal or complicated flow trajectories that make the sampling step expensive. An alternative approach to sampling in diffusion models involves solving the corresponding probability-flow ordinary differential equations (ODEs) (Song et al., 2020b; 2023). This has led to the development of faster samplers, such as DDIM (Song et al., 2020a), DPM solvers (Lu et al., 2022; Zheng et al., 2023), DEIS (Zhang & Chen, 2022), and Genie (Dockhorn et al., 2022). However, these methods still require dozens of inference steps to produce satisfactory results.

To alleviate this computational bottleneck in the sampling stage, (Liu et al., 2023b) recently proposed rectified flow, which aims to efficiently sample from the target distribution by iteratively learning the straight flow trajectories. To elucidate further, rectified flow starts from a potentially curved flow model, similar to DDIM (Song et al., 2020a) or other flow-based models (Lipman et al., 2022; Albergo & Vanden-Eijnden, 2022; Albergo et al., 2023), that transports the noise distribution to the target distribution, and then applies the reflow procedure to straighten the trajectories of the flow, thereby reducing the transport cost (Liu, 2022; Shaul et al., 2023b). Recent experimental studies in (Liu et al., 2024; 2023b) have demonstrated that rectified flow can achieve high-quality image generation within one or two steps just after 2-rectification procedures. (Lee et al., 2024) recently proposed an improved training routine for rectified flow and also achieved impressive results just after 2-rectification procedures. However, despite the computational advancements, a theoretical understanding of the convergence rate of rectified

flow to the true data distribution and the effect of straightness on its computational complexity remains elusive. In this paper, we investigate these issues and make the following contributions.

- *Wasserstein convergence and effect of straightness*: We establish a new bound for the squared 2-Wasserstein distance between the sampled data distribution in rectified flow and the true target distribution that mainly depends on the *estimation error* of the velocity (or drift) function and the *discretization error* induced by the Euler discretization scheme. Our upper bound is characterized by a novel *straightness parameters* of the flow that takes small values for near straight flows. Therefore, our result explains the rationale behind the sufficiency of fewer discretization steps in the sampling stage under near-straight flows with rigorous theoretical underpinning.

- *Straightness of 2-rectified flow*: We establish the *first theoretical result* to show that straight flows are provably achievable within only two rectification steps under mild regularity conditions. This result provides theoretical justification to the empirical finding, commonly encountered both in simulations and real-world data, that only two iterations of the RF procedure are often sufficient to produce straight flows. We also study the geometry of the flow when the source and target distributions are Gaussians and simple mixtures of Gaussians, respectively.

The rest of the paper is organized as follows: Section 2 provides some background on optimal transport and its connection with rectified flow. In Section 3, we present the main convergence results for the continuous time and discretized rectified flow under the 2-Wasserstein metric. We also introduce novel straightness parameters and study their effect on the convergence rate. Section 4 focuses on establishing a general straightness result for 2-RF under a very general setting and building geometric intuition for rectified flow under simpler but rather instructive examples. In particular, Section 4.1 provides a general result that shows the straightness of 2-RF between standard Gaussian and a target distribution within a fairly general class of distributions that also includes a general mixture of Gaussian distributions. Section 4.2 focuses on the geometry of 1-RF for some simpler Gaussian mixture models that also help to build a geometric intuition for the straightness phenomenon in 2-RF. In these special cases, we show that two rectifications are sufficient to obtain a straight flow. Finally, we present supporting simulated and real data experiments in Section 5 to empirically validate our theoretical findings.

**Notation.** Let $\mathbb{R}$ denote the set of real numbers. We denote by $\mathbb{R}^d$ the $d$-dimensional Euclidean space, and for a vector $x \in \mathbb{R}^d$, we denote by $\|x\|_2$ the $\ell_2$-norm of $x$. We use $I_d$ to denote the $d$-dimensional identity matrix. For a positive integer $K$, denote by $[K]$ the set $\{1, 2, \ldots, K\}$.

For a random variable $X$ we denote by $\mathrm{Law}(X)$ the probability distribution (or measure) of $X$. We write $X \sim \rho$ to denote $\rho = \mathrm{Law}(X)$. Moreover, for an absolutely continuous probability distribution $\rho$ with respect to the Lebesgue measure $\lambda$ over $\mathbb{R}^d$, we denote by $\frac{d\rho}{d\lambda}$ the Radon-Nikodym derivative of $\rho$ with respect to $\lambda$, i.e., the density of $X$ with respect to $\lambda$ is $\xi := \frac{d\rho}{d\lambda}$. For two distributions $\rho_1$ and $\rho_2$, we use $W_2(\rho_1, \rho_2)$ to denote the 2-Wasserstein distance between $\rho_1$ and $\rho_2$. $N(0, I_d)$ denotes the standard gaussian distribution in $\mathbb{R}^d$.

For a continuous and differentiable path $\{x_t\}_{t \in [0,1]} \subset \mathbb{R}^d$ and time varying functions $f_t : \mathbb{R}^d \to \mathbb{R}^m$, we denote by $\dot{f}_t(x_t)$ the time derivative of $f_t(x_t)$, i.e., $\dot{f}_t(x_t) = \frac{df_t(x_t)}{dt}$. Similarly, we use $\ddot{f}_t(x_t)$ to denote $\frac{d^2 f_t(x_t)}{dt^2}$. For a vector field $v : \mathbb{R}^d \to \mathbb{R}^d$, we let $\nabla \cdot v$ be its divergence.

Throughout the paper, we will use standard big-Oh (respectively big-Omega) notation. In detail, for a sequence $\{a_n\}$ of real numbers and a sequence $\{b_n\}$ of positive numbers, $a_n = O(b_n)$ (respectively $a_n = \Omega(b_n)$) signifies that there exists a universal constant $C > 0$, such that $|a_n| \leq Cb_n$ (respectively $|a_n| \geq Cb_n$) for all $n \in \mathbb{N}$.

## 2 BACKGROUND AND PRELIMINARIES

### 2.1 OPTIMAL TRANSPORT

The optimal transport (OT) problem in its original formulation as the Monge problem (Monge, 1781) is given by

$$\inf_{\mathcal{T}} \mathbb{E}\left[c(\mathcal{T}(X_0) - X_0)\right] \quad \text{s.t.} \quad \mathrm{Law}(\mathcal{T}(X_0)) = \rho_1, \quad \mathrm{Law}(X_0) = \rho_0,$$

where the the infimum is taken over deterministic couplings $(X_0, X_1)$ where $X_1 = \mathcal{T}(X_0)$ for $\mathcal{T} : \mathbb{R}^d \to \mathbb{R}^d$ to minimize the $c$-transport cost. See, e.g., Villani (2009). The Monge problem was relaxed by Kantorovich (Kantorovich, 1958) and the Monge-Kantorovich (MK) problem allowed for all (deterministic and stochastic) couplings

$(X_0, X_1)$ with marginal laws $\rho_0$ and $\rho_1$ respectively. However, it is well-known that if $\rho_0$ is an absolutely continuous probability measure on $\mathbb{R}^d$, both problems have the same optimal coupling that is deterministic, and hence, the optimization could be restricted only to the set of deterministic mappings $\mathcal{T}$. We consider an equivalent dynamic formulation of the Monge and MK problems as finding a continuous-time process $\{X_t\}_{t \in [0,1]}$ from the collection of all smooth interpolants $\mathcal{X}$ such that $X_0 \sim \rho_0$ and $X_1 \sim \rho_1$. For convex cost functions $c$, Jensen's inequality gives that

$$\mathbb{E}\left[c(X_1 - X_0)\right] = \mathbb{E}\left[c\left(\int_0^1 \dot{X}_t dt\right)\right] = \inf_{\{X_t\}_{t \in [0,1]} \in \mathcal{X}} \mathbb{E}\left[\int_0^1 c\left(\dot{X}_t\right) dt\right]$$

where the infimum is indeed achieved when $X_t = tX_1 + (1-t)X_0$, also known as the displacement interpolant, which forms a geodesic in the Wasserstein space (McCann, 1997). When we restrict the processes to those induced by the ODEs of the form $dX_t = v_t(X_t)dt$, the Lebesgue density of $X_t$, denoted by $\xi_t$, satisfies the continuity equation (also known as the Fokker-Planck equation) given by $\frac{\partial \xi_t}{\partial t} + \nabla \cdot (v_t \xi_t) = 0$, and the Monge problem can be recast as

$$\inf_{\{v_t\}_{t \in [0,1]}, \{X_t\}_{t \in [0,1]}} \mathbb{E}\left[\int_0^1 c\left(v_t(X_t)\right) dt\right], \quad \text{s.t.} \quad \frac{\partial \xi_t}{\partial t} + \nabla \cdot (v_t \xi_t) = 0, \quad \xi_0 = \frac{d\rho_0}{d\lambda} \quad \text{and} \quad \xi_1 = \frac{d\rho_1}{d\lambda}.$$

However, the dynamic formulation outlined above is challenging to solve in practice. When the cost function $c = \|\cdot\|^2$, this corresponds exactly to the kinetic energy objective introduced by (Shaul et al., 2023a), who demonstrate that the displacement interpolant minimizes the kinetic energy of the flow, resulting in straight-line flow paths. Additionally, (Liu, 2022) show that Rectified Flow, which iteratively learns the drift function $v_t$ for the displacement interpolant, simplifies this complex problem into a series of least-squares optimization tasks. With each iteration of Rectified Flow, the transport cost is reduced for all convex cost functions $c$.

## 2.2 RECTIFIED FLOW

In this section, we briefly introduce the basics of Rectified flow (Liu et al., 2023b; Liu, 2022), a generative model that transitions between two distributions $\rho_0$ and $\rho_1$ by solving ordinary differential equations (ODEs). Let $\rho_{\text{data}} := \rho_1 = \text{Law}(X_1)$ be the target data distribution on $\mathbb{R}^d$ and the linear-interpolation process be given by

$$X_t = tX_1 + (1-t)X_0, \quad 0 \le t \le 1$$

where $\rho_t = \text{Law}(X_t)$ and the starting distribution $\rho_0$ is typically a standard Gaussian or any other distribution that is easy to sample from. In the training phase, the procedure first learns the drift function $v : \mathbb{R}^d \times [0,1] \to \mathbb{R}^d$ as the solution to the optimization problem

$$v = \arg\min_f \int \mathbb{E}\left[\|\dot{X}_t - f(X_t, t)\|_2^2\right] dt = \arg\min_f \int \mathbb{E}\left[\|(X_1 - X_0) - f(X_t, t)\|_2^2\right] dt, \quad (1)$$

where the minimization is over all functions $f : \mathbb{R}^d \times [0,1] \to \mathbb{R}^d$. In practice, the initial coupling is usually an independent coupling, i.e., $(X_0, X_1) \sim \rho_0 \times \rho_1$. The MMSE objective in (1) is minimized at

$$v_t(x) := v(x, t) = \mathbb{E}\left[\dot{X}_t \mid X_t = x\right] = \mathbb{E}\left[X_1 - X_0 \mid X_t = x\right] \quad \text{for } t \in (0, 1). \quad (2)$$

For sampling, (Liu et al., 2023b) show that the ODE

$$dZ_t = v_t(Z_t) \, dt, \quad \text{where } Z_0 \sim \rho_0 \quad (3)$$

yields the same marginal distribution as $X_t$ for any $t$, i.e., $\text{Law}(Z_t) = \text{Law}(X_t) = \rho_t$, owing to the identical Fokker-Planck equations. We call $\mathcal{Z} = \{Z_t\}_{t \in [0,1]}$ the rectified flow of the coupling $(X_0, X_1)$, denoted as $\mathcal{Z} = \texttt{Rectflow}((X_0, X_1))$, and $(Z_0, Z_1)$ the rectified coupling, denoted as $(Z_0, Z_1) = \texttt{Rectify}((X_0, X_1))$.

The uniform Lipschitzness of the drift function $v_t$ for all $t \in [0,1]$ is a sufficient condition for the rectified flow $\mathcal{Z}$ to be unique (Murray & Miller, 2013, Theorem 1). Hence, the $\texttt{Rectflow}$ procedure rewires the trajectories of the linear interpolation process such that no two paths, corresponding to different initial conditions, intersect at the same time. After solving the ODE (3), one can also apply another $\texttt{Rectflow}$ procedure, also called Reflow or the 2-Rectified flow, to the coupling $(Z_0, Z_1)$ by learning the drift function

$$v_t^{(2)}(Z_t^{(2)}) = \mathbb{E}\left[Z_1 - Z_0 \mid tZ_1 + (1-t)Z_0 = Z_t^{(2)}\right].$$

This procedure can be done recursively, say $K$ times, resulting the $K$-Rectified Flow procedure. Liu et al. (2023b) shows that $K$-Rectified Flow couplings are straight in the limit of $K \to \infty$. We give the formal definition of straightness below.

**Definition 2.1.** *(Straight coupling and flow) A coupling $(X_0, X_1)$ is called straight or fully rectified when* $\mathbb{E}\left[X_1 - X_0 \mid tX_1 + (1-t)X_0\right] = X_1 - X_0$ *almost surely in* $t \sim \mathrm{Unif}([0,1])$.

*Moreover, for a straight coupling $(X_0, X_1)$, the corresponding rectified flow $\mathcal{Z} = \mathtt{Rectflow}((X_0, X_1))$ has straight line trajectories, and $(X_0, X_1) \stackrel{d}{=} (Z_0, Z_1) = \mathtt{Rectify}((X_0, X_1))$, i.e., they both have the same joint distribution. A flow that satisfies these properties is called a straight flow.*

Straight flows are especially appealing because, in practice, solving the ODE (3) analytically is rarely feasible, necessitating the use of discretization schemes for numerical solutions. However, for straight flows, the trajectories follow straight lines, allowing for closed-form solutions without the need for iterative numerical solvers, which significantly accelerates the sampling process.

Moreover, in practice, one is usually given samples from $\rho_{\mathrm{data}}$, and the drift function is estimated by empirically minimizing the objective in (1) over a large and expressive function class $\mathcal{F}$ (for example, the class of neural networks). Subsequently, the estimate $\widehat{v}_t$ is used to obtain the sampling ODE

$$d\tilde{Y}_t = \widehat{v}_t(\tilde{Y}_t)\,dt, \quad \text{where } \tilde{Y}_0 \sim N(0, I_d). \tag{4}$$

Because the solution to the ODE (4) is typically not analytically available, one must rely on discretization schemes. As proposed in Liu et al. (2023b), we apply the Euler discretization of the ODE to obtain our final sample estimates as mentioned below:

$$\widehat{Y}_{t_i} = \widehat{Y}_{t_{i-1}} + \widehat{v}_{t_{i-1}}(\widehat{Y}_{t_{i-1}})(t_i - t_{i-1}), \quad \text{for } i \in [T], \tag{5}$$

where the ODE is discretized into $T$ uniformly spaced steps, with $t_i = i/T$. The final sample estimate, $\widehat{Y}_1$, follows the distribution $\widehat{\rho}_{\mathrm{data}} := \mathrm{Law}(\widehat{Y}_1)$.

## 3 MAIN RESULTS ON WASSERSTEIN CONVERGENCE

### 3.1 CONTINUOUS TIME WASSERSTEIN CONVERGENCE

In this section, we study the convergence error rate of the final estimated distribution of the rectified flow. In particular, we establish error rates in the 2-Wasserstein distance for the estimated distributions procured through the approximate ODE flow (4). To this end, we make some useful assumptions on the drift function and its estimate that are necessary for establishing error bounds:

**Assumption 3.1.** *Assume that*

  (a) *(Estimation error) There exists an $\varepsilon_{\mathrm{vl}} \geq 0$ such that $\max_{0 \leq i \leq T} \mathbb{E}_{X_{t_i} \sim \rho_{t_i}} \|v_{t_i}(X_{t_i}) - \widehat{v}_{t_i}(X_{t_i})\|_2^2 \leq \varepsilon_{\mathrm{vl}}^2$.*

  (b) *(Lipschitz condition) The drift function $\widehat{v}_t$ satisfies $\|\widehat{v}_t(x) - \widehat{v}_t(y)\|_2 \leq \widehat{L}\,\|x - y\|_2$ almost surely, for some $\widehat{L} > 0$.*

Assumption 3.1(a) requires $\widehat{v}_t$ to be an accurate approximation of the original drift function $v_t$ for all the time points $t \in \{t_i\}_{i \in [T]}$. Assumptions of this nature are standard in diffusion model literature (Gupta et al., 2024; Li et al., 2024b;a; Chen et al., 2023), and they are indeed necessary to establish a reasonable bound on the error rate. Assumption 3.1(b) is a standard Lipschitz assumption on the estimated drift function $\widehat{v}_t$. In the literature concerning the score-based diffusion models and flow-based models, similar Lipschitzness (and *one-sided* Lipschitzness) assumptions on the estimated score functions of $\{X_{t_i}\}_{i \in [T]}$ are common (Chen et al., 2023; Kwon et al., 2022; Li et al., 2024b; Pedrotti et al., 2024; Boffi et al., 2024) requirement for theoretical analysis. In fact, $\widehat{v}$ is typically given by a neural network, which corresponds to a Lipschitz function for most practical activations. Moreover, in the context of rectified flow or flow-based generative models, the Lipschitzness condition on the true drift function $v_t$ is particularly an important requirement for the existence and uniqueness of the solution of the ODE (3) (Liu et al., 2023b; Boffi et al., 2024). Therefore, it is only natural to consider a class of neural networks that satisfies the Lipschitzness property for the training procedure.

Below, we present our first theorem, which bounds the error between the actual data distribution $\rho_1$ and the estimated distribution by following the exact ODE (4).

**Theorem 3.2.** *Let the condition of Assumption 3.1(b) hold, and also assume that $\rho_1$ is absolutely continuous with respect to the Lebesgue measure in $\mathbb{R}^d$. Also, write $b(t) = \mathbb{E}_{X_t \sim \rho_t} \|v_t(X_t) - \widehat{v}_t(X_t)\|_2^2$ for $t \in [0,1]$, and $\tilde{\rho}_1$ be*

*the distribution of $\tilde{Y}_1$. Then, almost surely,*

$$W_2^2(\tilde{\rho}_1, \rho_1) \le e^{1+2\widehat{L}} \int_0^1 b(t)\ dt.$$

The bound displayed in Theorem 3.2 is indeed very similar to the bounds obtained in Kwon et al. (2022); Pedrotti et al. (2024); Boffi et al. (2024), i.e., the bound essentially depends on the estimation error $b(t)$ for all $t \in [0,1]$. If there exists and $\varepsilon > 0$ such that $\sup_{t\in[0,1]} b(t) \le \varepsilon^2$, then we have the bound on the squared 2-Wasserstein to be of the order $O(\varepsilon^2)$. However, the requirement on $b(t)$ is much more stringent than Assumption 3.1(a) which amounts to bound on estimation error at the discrete time points $\{t_i\}_{i=0}^T$. The detailed proof is deferred to Appendix A.2.1. It is also worth mentioning that the Lipschitz assumption on $\widehat{v}$ can be relaxed to the one-sided Lipschitzness condition: if

$$\langle \widehat{v}_t(x) - \widehat{v}_t(y), x - y \rangle \le \widehat{L}\left\|x - y\right\|_2^2 \quad \text{for all } t \in [0,1],$$

almost surely, then the conclusions of Theorem 3.2 also hold true. Moreover, in this case, $\widehat{L}$ needs not be non-negative, as required in Assumption 3.1(b). Finally, unlike Chen et al. (2023); Gupta et al. (2024), we do not require any second-moment or sub-Gaussian assumption on $X_t$.

**Remark 1.** *The absolute continuity requirement in Theorem 3.2 can be relaxed. If the density of $\rho_1$ does not exist, then one can convolve $X_1$ with an independent noise $W_\eta \sim N(0, \eta I_d)$ for a very small $\eta > 0$, and consider the mollified distribution $\rho_1^\eta := \text{Law}(X + W_\eta)$ as the target distribution. Note that $\rho_1^\eta$ is absolutely continuous and satisfies $W_2^2(\rho_1^\eta, \rho_1) \le \eta^2 d$. Therefore, under the condition of Theorem 3.2, and using triangle inequality we have $W_2^2(\tilde{\rho}_1, \rho_1) \lesssim \eta^2 d + e^{1+2\widehat{L}} \int_0^1 b(t)\ dt$.*

## 3.2 STRAIGHTNESS AND WASSERSTEIN CONVERGENCE OF DISCRETIZED FLOW

In this section, we introduce a notion of straightness of the discretized flow (5), and study its effect on the Wasserstein convergence error rate between true data distribution $\rho_1$ and the sampled data distribution $\widehat{\rho}_{\text{data}}$. As we will see in the subsequent discussion, the straightness parameter of the ODE flow (3) plays an imperative role in the error rate, and our analysis shows that a more straight flow requires fewer discretization steps to achieve a reasonable error bound.

**New quantifiers for straightness of the flow.** We focus the ODE flow (3) assuming a standard Gaussian initial distribution, i.e.

$$dZ_t = v_t(Z_t)dt, \quad Z_0 \sim N(0, I_d).$$

Consider the random curve $\{\alpha(t)\}_{t\in[0,1]} \subset [0,1] \times \mathbb{R}^d$, where $\alpha(t) := (t, Z_t)$. The straightness of a twice-differential parametric curve determined by its curvature at each time point $t$, measured by the rate of change of the tangent vector $\dot{\alpha}(t) = (1, v_t(Z_t))$, which is essentially the acceleration of the particle at time $t$. To illustrate, consider the curve $\alpha(t) = (t, t)$, for $0 \le t \le 1$. The magnitude of the instantaneous acceleration is $\|\ddot{\alpha}(t)\|_2 = 0$. That is, $\alpha(t)$ has no curvature, i.e., it is straight. On the other hand, the curve given by $\alpha(t) = (\sin t, \cos t)$, for $0 \le t \le 1$, has (constant) curvature. Indeed the magnitude of the instantaneous acceleration is $\|\ddot{\alpha}(t)\|_2 = 1$ for all $t$.

The above discussion motivates us to define two key quantities to measure the straightness of the flow $\mathcal{Z}$:

**Definition 3.3.** *Let $\mathcal{Z} = \{Z_t\}_{t\in[0,1]}$ be twice-differentiable flow following the ODE (3).*

1. *The average straightness (AS) parameter of $\mathcal{Z}$ is defined as*

$$\gamma_1(\mathcal{Z}) := \int_0^1 \mathbb{E}\left\|\dot{v}_t(Z_t)\right\|_2^2\ dt.$$

2. *Let $0 = t_0 < t_1 < \ldots < t_T = 1$ be a partition of $[0,1]$ into $T$ intervals of equal length. The piece-wise straightness (PWS) parameter of the flow $\mathcal{Z}$ is defined as*

$$\gamma_{2,T}(\mathcal{Z}) := \max_{i\in[T]} \frac{1}{t_i - t_{i-1}} \int_{t_{i-1}}^{t_i} \mathbb{E}\left\|\dot{v}_t(Z_t)\right\|_2^2\ dt.$$

The quantity $\gamma_1(\mathcal{Z})$ essentially captures the average straightness of the flow along the time $t \in [0,1]$. On the other hand, $\gamma_{2,T}(\mathcal{Z})$ captures the degree of straightness of $\mathcal{Z}$ for every interval $[t_{i-1}, t_t]$ for all $i \in [T]$. Therefore, $\gamma_{2,T}(\mathcal{Z})$ captures a somewhat more stringent notion of straightness. In addition, a small value of $\gamma_1(\mathcal{Z})$ or $\gamma_{2,T}(\mathcal{Z})$ indicates that the flow is close to perfect straightness. In fact, $\gamma_1(\mathcal{Z}) = 0$ or $\gamma_{2,T}(\mathcal{Z}) = 0$ implies that the flow $\mathcal{Z}$ is a *straight flow* in the sense of Definition 2.1. To formally state the claim, let

$$S(\mathcal{Z}) := \int_0^1 \mathbb{E} \left\| Z_1 - Z_0 - v_t(Z_t) \right\|_2^2 \, dt.$$

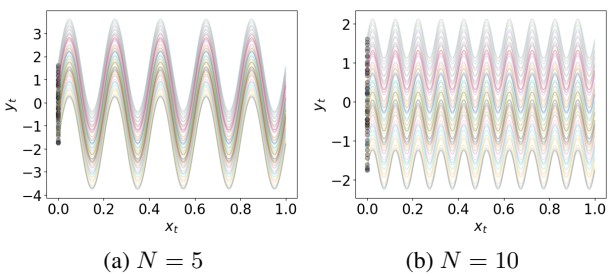

(a) $N = 5$  (b) $N = 10$

Figure 1: Flow of $Z_t = Z_0 + (t, 50N^{-2}\sin(2\pi Nt))^\top$ for different choices of $N$.

This quantity was introduced in Liu et al. (2023b) to quantify the degree of straightness of the flow $\mathcal{Z}$. Specifically, Liu et al. (2023b) showed that $S(\mathcal{Z}) = 0$ if and only if $\mathcal{Z}$ is a straight flow. The next lemma compares the above notions of straightness.

**Lemma 3.4.** *The AS and PWS parameters satisfy $\gamma_{2,T}(\mathcal{Z}) \geq \gamma_1(\mathcal{Z}) \geq S(\mathcal{Z})$. Moreover, $S(\mathcal{Z}) = 0$ if and only if $\gamma_1(\mathcal{Z}) = \gamma_{2,T}(\mathcal{Z}) = 0$.*

The above lemma tells that a flow which is a near-straight flow in the notion of AS or PWS (i.e. $\gamma_1(\mathcal{Z})$ and $\gamma_{2,T}(\mathcal{Z})$ are small), is also near-straight flow in terms of $S(\mathcal{Z})$. Moreover, the second part of the above lemma shows that $\gamma_1(\mathcal{Z}) = 0$ iff $\mathcal{Z}$ is *straight*, i.e, the notion of a *perfectly straight flow* in terms of AS aligns with that of a straight flow of Liu et al. (2023b).

However, we argue that $S(\mathcal{Z})$ could lead to a misleading notion of *near-straightness* that may conflict with our intuitive perception of a near-straight flow. To elaborate, a flow $\mathcal{Z}$ could exist such that $S(\mathcal{Z})$ could be close to zero but $\gamma_1(\mathcal{Z})$ is well bounded away from 0. We illustrate this phenomenon through the following examples.

**Example 1.** *Consider the velocity function $v_t(Z_t) = \frac{1}{2\pi N}(\sin(2\pi Nt), \cos(2\pi Nt))^\top$, where $N \in \mathbb{N}$ and $t \in [0,1]$. The path of the flow is a circle. In this case, $S(\mathcal{Z}) = O(N^{-2}) \to 0$ as $N \to \infty$. Therefore, $S(\mathcal{Z})$ clearly fails to capture the degree of curvature of $\mathcal{Z}$ for large $N$. However, $\gamma_{2,T}(\mathcal{Z}) = \gamma_1(\mathcal{Z}) = 1$, i.e., AS and PWS are able to capture the departure of $\mathcal{Z}$ from straightness.*

**Example 2.** *Let $N \in \mathbb{N}$ and consider the ODE flow (3) with $v_t(Z_t) = (1, 100\pi N^{-1}\cos(2\pi Nt))^\top$. In this case, we have $Z_t = Z_0 + (t, 50N^{-2}\sin(2\pi Nt))^\top$ and $\dot{v}_t(Z_t) = -(0, 200\pi^2 \sin(2\pi Nt))^\top$. Straightforward calculations show that $S(\mathcal{Z}) = O(N^{-2})$, while $\gamma_{2,T}(\mathcal{Z}) \geq \gamma_1(\mathcal{Z}) = 2 \times 10^4 \pi^4$. Therefore, $S(\mathcal{Z})$ can be arbitrarily close to 0 as $N \to \infty$, whereas $\gamma_1(\mathcal{Z})$ and $\gamma_{2,T}(\mathcal{Z})$ remain bounded away from zero. We also observe in Figure 1 that the undulation of the flow is greater for $N = 10$ compared to $N = 5$, i.e. the curvature increase with $N$.*

We are now ready to state our main result about Wasserstein convergence for the discretized ODE (5).

**Theorem 3.5.** *Let Assumption 3.1 hold for the flow $\mathcal{Z} := \{Z_t\}_{0 \leq t \leq 1}$ determined by the ODE (3), assuming a differentiable velocity field $v : \mathbb{R}^d \times [0,1] \to \mathbb{R}^d$. Then the estimate of the distribution $\widehat{\rho}_{\text{data}}$ obtained through the ODE (5) satisfies the following almost sure inequality:*

$$W_2^2(\widehat{\rho}_{\text{data}}, \rho_1) \leq \frac{27 e^{4\widehat{L}}}{\max\{\widehat{L}^2, 1\}} \left( \frac{\gamma_{2,T}(\mathcal{Z})}{T^2} + \varepsilon_{\text{vl}}^2 \right),$$

The term involving the PWS parameters could be referred to as an error term due to discretization. More importantly, the above Wasserstein error bound shows that $T = \Omega\left(\sqrt{\gamma_{2,T}(\mathcal{Z})/\epsilon}\right)$ is sufficient to achieve a discretization error of the order $O(\epsilon)$. Therefore, Theorem 3.5 indicates that if the flow is a near-straight flow (i.e., $\gamma_{2,T}(\mathcal{Z}) \approx 0$), then accurate estimation of the data distribution can be achieved with a very few discretization steps. This phenomenon indeed aligns with the empirical findings in Liu et al. (2023b); Lee et al. (2024); Liu et al. (2024) related to the rectified flow. To further elaborate, Theorem 3.5 shows that if a flow enjoys better *piece-wise straightness* in each partitioning interval, we need fewer discretization steps to achieve desirable accuracy compared to the case of a flow that deviates from straightness. This is also consistent with the empirical behavior of Perflow (Yan et al., 2024), a methodology that has achieved state-of-the-art performance by further straightening the rectified flow in each interval $[t_{i-1}, t_i]$ for all $i \in [T]$. The proof of the theorem can be found in Appendix A.2.3. It is also

worthwhile to point out that one can obtain a Wasserstein error bound using the AS parameter since this relates the error rate to the average notion of straightness that could be useful for practical purposes as it does not depend on the coarseness of the partition. To this end, we have the elementary inequality $\gamma_{2,T}(\mathcal{Z}) \leq T\gamma_1(\mathcal{Z})$ (see Appendix A.2.2) which immediately leads to the following corollary.

**Corollary 3.6.** *Under the same conditions of Theorem 3.5, we have the following almost sure inequality:*

$$W_2^2(\widehat{\rho}_{\text{data}}, \rho_1) \leq \frac{27e^{4\widehat{L}}}{\max\{\widehat{L}^2, 1\}} \left( \frac{\gamma_1(\mathcal{Z})}{T} + \varepsilon_{\text{vl}}^2 \right).$$

## 4 ONE RECTIFICATION LEADS TO STRAIGHT COUPLING IN MOST CASES

Although the trajectories of 1-Rectified flow are non-intersecting (because the drift function is Lipschitz continuous), the algorithm is not guaranteed to return a straight flow, potentially requiring a large number of discretization steps (or drift function evaluations) to generate high-quality samples. Liu et al. (2023b) show that repeatedly applying the Rectified Flow procedure progressiveness reduces the curvature of the flow, producing a straight flow in the limit as the number $K$ of iteration in $K$-Rectified Flow increases. Liu (2022); Liu et al. (2024) empirically show that one needs at least three applications of the rectified flow for a fair one-step generation quality. On the other hand, (Lee et al., 2024) heuristically suggests that no more than two applications are required, though a formal theoretical justification remains unproven. In this section, we will show that, under some mild regularity conditions, 1-Rectified Flow (1-RF) yields straight coupling (or 2-RF generates straight flow) between the standard Gaussian distribution and a fairly broad class of target distributions that also includes general mixtures of Gaussians, thus providing theoretical underpinning to the numerical findings in prior literature.

### 4.1 A GENERAL RESULT FOR STRAIGHTNESS

In Section 3, we assumed that the learned velocities $\widehat{v}_t$ are Lipschitz functions, and argued that global Lipschitzness is sufficient for the existence of a unique solution to ODE (3). However, such conditions might be a bit too strong, even in some simple cases. In fact, when $X_1$ follows a general mixture of Gaussian distribution, the global Lipschitz condition may not hold or hold with a very large constant. In this section, we will work with somewhat more pragmatic conditions on the true velocity functions $v_t$. Consider the non-stochastic version of ODE (3), i.e.,

$$dZ_t = v_t(Z_t)\,dt, \quad Z_0 = z_0. \tag{6}$$

For clarity, we denote the solution of the above ODE as $Z_t(z_0)$ in contrast to the solution $Z_t$ of the ODE (3), which has a random starting point.

**Definition 4.1.** *For a positive integer $k$, a function $f : \mathbb{R}^d \to \mathbb{R}^d$ is said to be $\mathcal{C}^k$ if it is $k$-times continuously differentiable. Additionally, $f$ is called a $\mathcal{C}^{1,1}$ function if $f$ is a $\mathcal{C}^1$ function and its Jacobian is locally Lipschitz, i.e., for every $x \in \mathbb{R}^d$, there exists $\delta > 0$ and $L_{loc} > 0$ (which may depend on $x$) such that*

$$\max\{\|x - x_1\|_2, \|x - x_2\|_2\} \leq \delta \Rightarrow \|\nabla_x f(x_1) - \nabla_x f(x_2)\|_{op} \leq L_{loc} \|x_1 - x_2\|_2.$$

**Assumption 4.2.** *We assume that the velocity function $v_t(\cdot)$ is a $\mathcal{C}^{1,1}$ function for all $t \in [0, 1]$.*

Note that, if $v_t(\cdot)$ is a $\mathcal{C}^2$ function, then it automatically satisfies Assumption 4.2. Therefore, global Lipschitzness is not required for the above assumption. However, Assumption 4.2 is not necessarily a weaker assumption as a Lipschitz function might not be a $\mathcal{C}^{1,1}$ function. Now we present a general result on the straightness of Rectified Flow.

**Theorem 4.3.** *Let $\mathbb{E}\|X_1\|_2 < \infty$ and the Assumption 4.2 hold. Also, assume that the solution to the ODE (6) satisfies the non-explosive condition*

$$\sup_{t \in [0,1]} \|Z_t(z_0)\|_2 < \infty \quad \text{for all initial values } z_0 \in \mathbb{R}^d. \tag{7}$$

*Then the rectified coupling $(Z_0, Z_1) := \texttt{Rectify}(X_0, X_1)$ is a straight coupling.*

*Proof sketch.* The main step in the proof is to show that the map $H_t(z_0) := (1 - t)z_0 + tZ_1(z_0)$ is almost everywhere locally invertible: this will lead to the straightness condition in Definition 2.1. Toward that goal, we deploy the inverse function theorem (IVT). First, borrowing tools from (Kunita, 1984), we establish the *existence*

*and uniqueness* of $\nabla_{z_0} Z_1(z_0)$ under Assumption 4.2 and Condition (7). Finally, we show that $\nabla_{z_0} H_t(z_0)$ is invertible for all $t$ outside a countable subset of $[0, 1]$, i.e., $\nabla_{z_0} H_t(z_0)$ is invertible almost surely in $t \sim \mathrm{Unif}([0, 1])$. This allows us to apply the IVT along with a careful covering argument for almost all $t$ (see Appendix A.3.1) and establish our claim.

**Non-explosivity.** Now, we provide a sufficient condition for non-explosivity that is easier to check so that Theorem 4.3 can be of practical use.

**Assumption 4.4** (Osgood type criterion (Osgood, 1898; Groisman & Rossi, 2007))**.** *Let $Z_t(z_0) \in \mathbb{R}^d$ be the solution of the ODE* (6), *where $(z_0, t) \in \mathbb{R}^d \times [0, 1]$. There exists a non-negative locally-Lipschitz (or strictly increasing) function $h : \mathbb{R}_+ \to \mathbb{R}_+$ such that*

$$\int_{u_0}^{\infty} \frac{1}{h(u)} \, du > 1, \quad \text{for all } u_0 > 0, \tag{8}$$

*and $\langle Z_t(z_0), v_t(Z_t(z_0)) \rangle \leq h(\|Z_t(z_0)\|_2^2)$, for all $(z_0, t) \in \mathbb{R}^d \times [0, 1]$.*

One sufficient condition is that $\sup_{t \in [0,1]} \langle x, v_t(x) \rangle \leq h(\|x\|_2^2)$ for all $x \in \mathbb{R}^d$ and for a positive locally-Lipschitz (or strictly increasing) function $h$ satisfying (8). The above criterion ensures that $\|Z_t(z_0)\|_2$ is always finite for all $t \in [0, 1]$ (Groisman & Rossi, 2007), i.e., the solutions does not explode. To be precise, the integral in (8) quantizes the explosion time of $\|Z_t(z_0)\|_2$, and it ensures that the explosion time falls outside $[0, 1]$. Moreover, as opposed to condition (7), this can be easily checked for a large class of target distributions, e.g., a general mixture of Gaussians. For example, for $(X_0, X_1) \sim N(0, I_d) \times \rho_1$ with $\rho_1 = \sum_{j=1}^{J} \pi_j N(\mu_j, \Sigma_j)$, it follows that $\sup_{t \in [0,1]} \langle x, v_t(x) \rangle \leq A \|x\|_2^2 + B \|x\|_2$ for some $A, B > 0$ (see Appendix A.3.3). Therefore, $h(u) = Au + B\sqrt{u}$ is a valid choice and it also satisfies Assumption 4.4, as $\int_{u_0}^{\infty} (Au + B\sqrt{u})^{-1} \, du = \infty$ for all $u_0 > 0$.

We are now ready to state the main result of this section.

**Theorem 4.5.** *Let $(X_0, X_1) \sim N(0, I_d) \times \rho_1$ such that $\mathbb{E} \|X_1\|_2 < \infty$. Also, let the condition in Assumption 4.4 hold for ODE* (6). *Then, the resulting rectified coupling $(Z_0, Z_1) := \mathtt{Rectify}(X_0, X_1)$ is a straight coupling.*

The above theorem gives a fairly general straightness guarantee for Rectified Flow starting from an independent coupling that covers a large class of target distributions. Essentially, the first moment ensures that Assumption 4.2 is satisfied. Therefore, coupled with Assumption 4.4, the conditions of Theorem 4.3 are satisfied, and hence, straightness follows. As a result, when $\rho_1$ is a general mixture of Gaussian, 1-RF yields straight coupling. The complete proof is deferred to Appendix A.3.2. To the best of our knowledge, Theorem 4.5 is the first result demonstrating that 1-RF produces a straight coupling with mild regularity assumptions. This provides concrete theoretical support for empirical findings in the prior works.

## 4.2 Examples with simple Gaussian mixtures

Although Theorem 4.5 is a quite strong result, it does not shed any light on the exact form or the geometry of the rectified flow. Therefore, in this section, we provide simple examples of RF for mixtures of Gaussians to elucidate its geometrical aspects. While simple, these examples provide intuition and further insights into understanding the straightness of rectified flow. We begin with the case of $\rho_0 = N(0, I_d)$ and $\rho_1 = N(\mu, \Sigma)$. We show that the 1-Rectified flow obtains the *optimal transport mapping* (with respect to the squared distance cost function) and is straight. The proof is deferred to Appendix A.3.4.

**Theorem 4.6.** *Let $(X_0, X_1) \sim \rho_0 \times \rho_1$ be an independent coupling where $\rho_0 = N(0, I_d)$ and $\rho_1 = N(\mu, \Sigma)$ where $\mu \in \mathbb{R}^d$, and $\Sigma$ is a $d \times d$ positive semi-definite matrix. The associated rectified coupling $(Z_0, Z_1) = \mathtt{Rectify}((X_0, X_1))$ is an optimal solution to the Monge problem, i.e., it minimizes $\mathbb{E}\left[c(Z_0 - \mathcal{T}(Z_0))\right]$ where $\mathcal{T}(Z_0) = Z_1$ amongst all deterministic couplings $\mathcal{T}$, for $c = \|\cdot\|_2^2$. Moreover, the coupling is given by $Z_1 = \Sigma^{1/2} Z_0 + \mu$.*

The above theorem shows that for a simple Gaussian to Gaussian case, 1-RF generates a straight coupling and solves the Monge problem. Moreover, it provides the exact form of the coupling. The rest of the section considers target distributions that are multimodal. Consider a simple case of $\rho_0 = N(0, 1)$ and $\rho_1 = .5N(y, 1) + .5N(-y, 1)$. It turns out that in this case, the flow induced by $v_t$ has an interesting geometric structure (see, for example, Figure A.1 (b)). In particular, if $z_0$ is positive (negative), then $z_t := Z_t(z_0)$ is also positive (negative) *for all $t$*. This follows from a very fundamental fact. First, note that 1-RF generates a straight coupling (Theorem 4.5) in

this case which is also monotonically increasing, i.e., for $(z_0, z_1)$ and $(z_0', z_1')$ such that $z_0 < z_0'$, we must have that $z_1 < z_1'$. We formalize this idea in the following lemma borrowed from (Liu et al., 2023b).

**Lemma 4.7** (Lemma D.9; (Liu et al., 2023b))**.** *A rectified coupling in $\mathbb{R}$ is straight iff it is deterministic and monotonic.*

In fact, the $v_t$ function is Lipschitz in this case which further ensures that the solution to ODE (6) is unique. This actually ensures that order is preserved for all $\{z_t\}_{t \in [0,1]}$ in the aforementioned example. This is a simple consequence of the Picard-Lindelof theorem, and the detailed proof is given in Appendix A.3.6 (Lemma A.2). We generalize this phenomenon in Lemma 4.8, which shows that in one dimension, the map $z_0 \mapsto z_t$ preserves the quantiles for all $t \in [0, 1]$.

**Lemma 4.8.** *Let $z_0 \in \mathbb{R}$ and write $z_t := Z_t(z_0)$. If the drift function $v_t(x)$ in ODE (3) is Lipschitz, then $\mathbb{P}(Z_t \leq z_t)$ is a constant depending on $z_0$ for all $t$.*

The detailed proof is deferred to Appendix A.3.5, and additional experiments can be found in Appendix A.1.1. This now paves the way for our next results.

**Proposition 4.9.** *Consider $(X_0, X_1) \sim \rho_0 \times \rho_1$, where $\rho_1 = \pi N(\mu_1, I_d) + (1 - \pi)N(\mu_2, I_d)$ where $\mu_1, \mu_2 \in \mathbb{R}^d$, and $\rho_0 = N(0, I_d)$. Then, 1-RF yields a straight coupling.*

*Proof sketch:* The result is a direct consequence of Theorem 4.5. However, we present a more intuitive and instructive proof sketch here. Our proof (Appendix A.3.7) proceeds by using a rotation to reduce the $d$ dimensional target distribution into another where the means of the two components of the Gaussian mixture are sparse with two non-zero coefficients, one of which is equal (lets say coordinate 1). We then show that the ODE decouples the flow and it can be analyzed coordinate-wise. Then we use Lemma 4.7 for the coordinates to prove straightness.

Finally, we come to the Gaussian mixture to Gaussian mixture setting.

**Proposition 4.10.** *Consider $\mu_{01} = (0, a)^\top, \mu_{02} = (0, -a)^\top$ and $\mu_{11} = (a, a)^\top, \mu_{12} = (a, -a)^\top$ for some $a > 0$. Let $X_0 \sim 0.5N(\mu_{01}, I_2) + 0.5N(\mu_{02}, I_2)$ and $X_1 \sim 0.5N(\mu_{11}, I_2) + 0.5N(\mu_{12}, I_2)$. Also, assume that $X_0, X_1$ are independent. Then 1-RF yields a straight coupling.*

The intuitive explanation is that even in this case, the flows along each coordinate decouples. The $x-$coordinate goes through a translation, whereas the $y-$coordinate's velocity function is uniformly Lipschitz, leading to a monotonic coupling along $y-$direction. This, along with Lemma 4.7, shows that the flow along the $y-$axis is also monotonic; hence, one rectification gives a straight coupling. The proof is deferred to the Appendix A.3.8.

## 5 EXPERIMENTS

In this section, we present numerical experiments for both synthetic and real data. We primarily explore the effect of the number of discretization steps $T$ and the straightness parameter $\gamma_{2,T}(\mathcal{Z})$ on the $W_2$ distance between the target distribution and the distribution of the generated samples after 1-rectification. Additional experiments can be found in Appendix A.1.

**Synthetic data.** For simulated data, we consider the following two examples: 1) Flow from standard Gaussian to a *balanced* mixture of Gaussian distributions in $\mathbb{R}^2$ with varying components, and 2) Flow from standard Gaussian to a checker-board distribution (see Figure A.2) with varying components. We detail our findings for the Gaussian mixture below and defer the Checkerboard example to Appendix A.1.2.

We choose the target distribution to be mixture of Gaussians with equal cluster probability and unit variance, where the number of components $K$ varies within $\{1, 2, 3, 4\}$. In all the cases, we show that the actual Wasserstein error is closely characterized by $\gamma_{2,T}$. To this end, we choose the all the means to have equal norm and similar separation as they both affect the Lipschitz constant of the drift function[1], which directly impacts the Wasserstein error as shown in our analysis (see Theorem 3.5.) For $K = 1$, we set mean of the target distribution to be $\mu_1 = (5, 0)^\top$; for $K = 2$: $\mu_1 = (5, 0)^\top, \mu_2 = (0, 5)^\top$; for $K = 3$: $\mu_1 = (5, 0)^\top, \mu_2 = (0, 5)^\top, \mu_3 = (-5, 0)^\top$; for $K = 4$: $\mu_1 = (-\frac{5}{\sqrt{2}}, \frac{5}{\sqrt{2}})^\top, \mu_2 = (5/\sqrt{2}, -5/\sqrt{2})^\top, \mu_3 = (-5/\sqrt{2}, -5/\sqrt{2})^\top, \mu_4 = (5/\sqrt{2}, 5/\sqrt{2})^\top$.

We start with the independent coupling in each of the four cases and and train a feed-forward neural network to estimate the drift function and generate the 1-rectified flow. Figure 2(a) shows that $W_2(\widehat{\rho}_{\text{data}}, \rho_1)$ decreases with

---

[1]An upper bound on the Lipschitz constant for a Gaussian mixture is given by $L \leq 2(1 + D \times R)$, where $D = \max_i \|\mu_i\|$ and $R = \max_{i,j} \|\mu_i - \mu_j\|$.

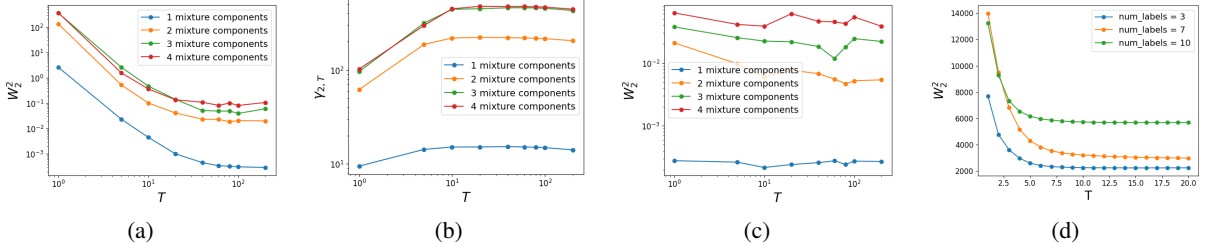

Figure 2: (a) $W_2^2(\widehat{\rho}_{\mathrm{data}}, \rho_1)$ vs $T$ (in log-log scale) for mixtures of Gaussians with varying components. (b) The straightness parameter $\gamma_{2,T}(\mathcal{Z})$ vs $T$ (in log-log scale) for the same respective distributions. (c) $W_2^2(\widehat{\rho}_{\mathrm{data}}, \rho_1)$ vs $T$ (in log-log scale) for the second rectification on the Gaussian mixtures (d) shows the $W_2^2(\widehat{\rho}_{\mathrm{data}}, \rho_1)$ vs $T$ for the FashionMNIST dataset with varying components. We observe that the straightness of the flow decreases with increasing number of mixture components.

increasing number of discretization steps $T$, and the slope of the curve is approximately 2 (before it stabilizes), which indeed validates the $1/T^2$ dependence that we show in Theorem 3.5. Moreover, $W_2$ distance is consistently larger for the flow corresponding to a larger number of components, owing to a larger value of the straightness parameter $\gamma_{2,T}(\mathcal{Z})$ as shown in Figure 2(b). Moreover, Figure 2(c), further validates our claim that the second rectified flow for Gaussian mixtures produces a straight flow– the Wasserstein error even with a single discretization step is close to 0.

**Real data.** For the real data experiments, we consider the MNIST and FashionMNIST datasets. In both examples, we train a UNet architecture-based network on training data to estimate the drift function and then evaluate the Wasserstein distance of the generated samples from the test split of the data. We give details for the FashionMNIST dataset here and defer MNIST to Appendix A.1.3. To emulate the behavior of having different number of modes, we consider three subsets of the FashionMNIST dataset consisting of the first 3 labels, the first 7 labels, and all 10 labels. We observe in Figure 2(d) that similar to the Gaussian mixture example, the presence of a higher number of components negatively affects the Wasserstein distance, again indicating that the flow becomes less straight with the increasing number of modes.

## 6 DISCUSSION AND FUTURE WORKS

Rectified Flow, a newly introduced alternative to diffusion models, is known to enjoy a fast generation time due to its ability to learn straight flow trajectories from noise to data. Existing works have empirically shown that 1-RF produces a *straight coupling* for many target distributions. To our knowledge, this paper is the first to show that this is indeed true for a large class of source and target distributions that are "nice". We also provide the first analysis of the Wasserstein distance between the sampling distribution of RF and the target distribution that connects the error rate with the straightness of the flow along with supporting experiments on real and simulated datasets.

Our analysis poses the natural question: are there source and target distributions where 1-RF does not give a straight coupling? The simple examples we came up with are those for which the Monge map (or any deterministic coupling) does not exist. However, under the knowledge of Theorem 4.3, we conjecture that an even more general version of Theorem 4.5 is possible. We suspect that if $(X_0, X_1)$ is rectifiable for some initial choices of distributions, then the 1-RF flow will result in a straight coupling under very mild regularity conditions. In fact, we conjecture that 1-RF (under certain regularity conditions) will result in the optimal coupling induced by the Monge map (if it exists), as it is known that RF iteratively solves the OT problem (Liu, 2022). Although this is an interesting research direction, we defer it to future research. Another direction of research could be to improve the dependence of the Lipschitz constant $\widehat{L}$ in Theorem 3.5, or theoretically explore the generalization error described in Assumption 3.1(a).

**Acknowledgement:** We thank Dr. Dheeraj Nagaraj at Google DeepMind, Bangalore for helpful discussions and for spotting an error in the initial proof of Theorem 4.3. We are also grateful to the reviewers whose suggestions helped improve the paper. PS gratefully acknowledges NSF grants 2217069, 2019844, and DMS 2109155.

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

APPENDIX

## A.1 MORE EXPERIMENTS

### A.1.1 EXPERIMENTS RELATED TO LEMMA 4.8

In Figure A.1 (a) we show the trajectories $(t, Z_t)$ of $n = 101$ data points generated from source distribution $\rho_0 = N(0, 1)$ and target distribution $\rho_1 = .5N(4, 1) + .5N(-10, 1)$. The blue, red, and black lines indicate the trajectory of the maximum, median, and minimum of the source samples, and the triangles indicate the maximum, median, and minimum of $n$ data points over time. One can see that the image of the same point at time $t$ continues to preserve the quantiles for all $t \in [0, 1]$. This phenomenon also leads to interesting geometrical phenomena. For example, Figure A.1 (b) shows that for transforming a Gaussian to a symmetric two-component mixture of Gaussians $.5N(10, 1) + .5N(-10, 1)$, all points above (below) the $Z_t = 0$ line stay above (below).

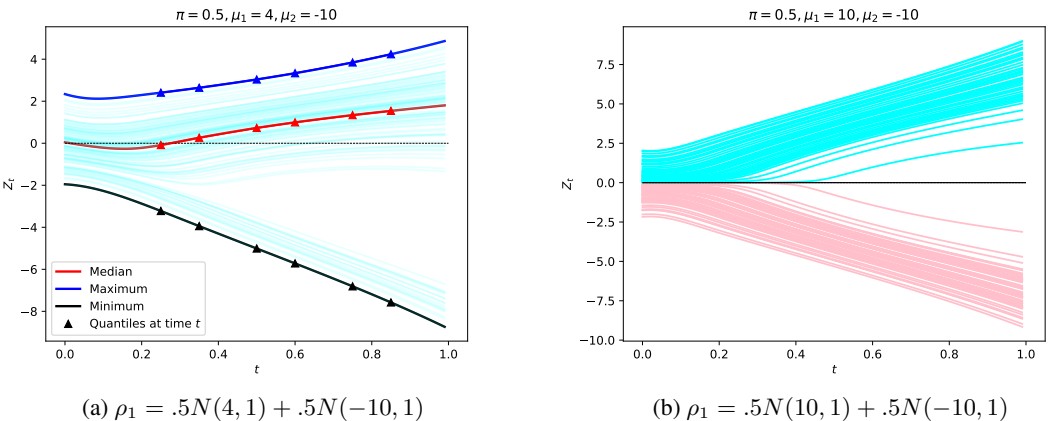

(a) $\rho_1 = .5N(4, 1) + .5N(-10, 1)$     (b) $\rho_1 = .5N(10, 1) + .5N(-10, 1)$

Figure A.1: (a) shows the flow of the minimum, median, and maximum values of a set of points, initially distributed according to a standard Gaussian. (b) shows the flow of points from a standard Gaussian to a symmetric mixture of two Gaussians and the black line represents $y = 0$.

### A.1.2 CHECKCER BOARD EXAMPLE

We consider the checker-board distribution with $2, 5$ and $8$ components. We use training datasets of size 10,000 to train a feed-forward neural network in order to learn the velocity drift function and evaluate $W_2^2(\widehat{\rho}_{\text{data}}, \rho_1)$ using POT (Feydy et al., 2019) for different levels of discretization $T$ over test data of size 5000. Figure A.2(d) also shows that larger component size has a negative effect on the Wasserstein distance, which stems from the fact that a larger number of components typically pushes the flow away from straightness.

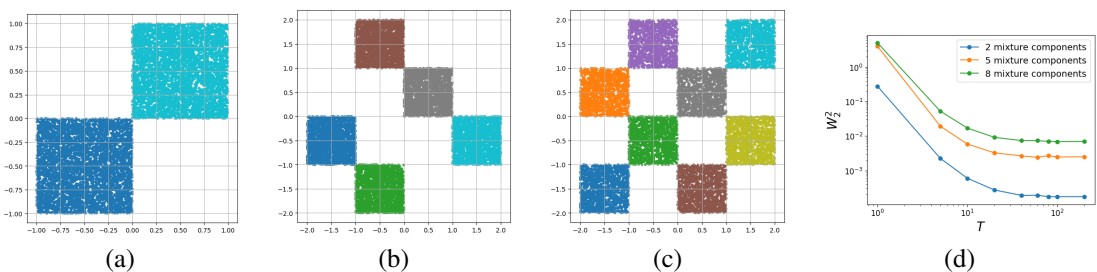

(a)     (b)     (c)     (d)

Figure A.2: (a) Checker-board distribution with 2 components. (b) Checker-board distribution with 5 components. (c) Checker-board distribution with 8 components. (d) shows the $W_2^2(\widehat{\rho}_{\text{data}}, \rho_1)$ vs $T$ (on log-log scale) for the FashionMNIST dataset with varying components..

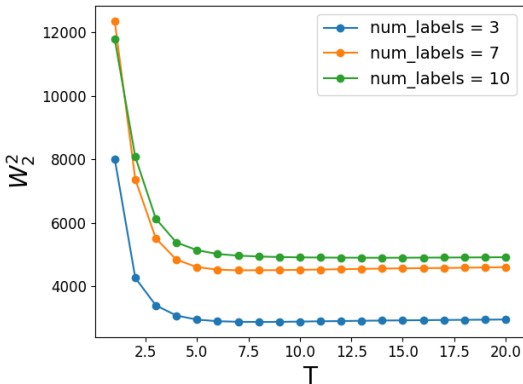

Figure A.3: shows the plot of squared Wasserstein distance $W_2^2(\widehat{\rho}_{\text{data}}, \rho_1)$ vs $T$ for the MNIST dataset with a varying numbers of labels, and hence, conditional modes.

### A.1.3 MNIST DATASET EXPERIMENT

For MNIST data, we construct 3-different datasets. The first one only contains the digits $\{0, 1, 2\}$, the second one only contains $\{0, 1, 2, \dots, 6\}$ and the final one contains $\{0, 1, 2, \dots, 9\}$. Essentially, these datasets contain multiple modes which resembles the nature of the synthetic dataset examples discussed in the previous section. Figure A.3(a) shows that the Wasserstein distance is larger when there is more number of components in the dataset. Essentially, more components make the flow more non-straight, and hence convergence in Wasserstein is affected.

## A.2 PROOFS OF SECTION 3

### A.2.1 PROOF OF THEOREM 3.2

Let $\{\rho_t\}_{t\in[0,1]}$ and $\{\tilde{\rho}_t\}_{t\in[0,1]}$ be distribution of the solution of (3) and (4) respectively. Let $\pi_t$ be the optimal coupling between $\rho_t$ and $\tilde{\rho}_t$. Therefore, using Corollary 5.25 of Santambrogio (2015), we have

$$
\begin{aligned}
\frac{1}{2}\frac{dW_2^2(\rho_t, \tilde{\rho}_t)}{dt} &= \int \langle x - y, v_t(x) - \widehat{v}_t(y) \rangle \; d\pi_t(x, y) \\
&= \int \langle x - y, v_t(x) - \widehat{v}_t(x) \rangle \; d\pi_t(x, y) + \int \langle x - y, \widehat{v}_t(x) - \widehat{v}_t(y) \rangle \; d\pi_t(x, y) \\
&\leq \frac{1}{2} \int \|x - y\|_2^2 \; d\pi_t(x, y) + \frac{1}{2} \int \|v_t(x) - \widehat{v}_t(x)\|_2^2 \; d\pi_t(x, y) + \widehat{L} \int \|x - y\|_2^2 \; d\pi_t(x, y) \\
&= (1/2 + \widehat{L}) W_2^2(\rho_t, \tilde{\rho}_t) + \frac{b(t)}{2}.
\end{aligned}
$$

Solving the above differential inequality leads to the following inequality

$$
W_2^2(\rho_\tau, \tilde{\rho}_\tau) \leq W_2^2(\rho_0, \tilde{\rho}_0) + e^{1+2\widehat{L}} \int_0^\tau b(t) \; dt.
$$

The result follows by noting that $W_2^2(\rho_0, \tilde{\rho}_0) = 0$ and setting $\tau = 1$.

A.2.2  PROOF OF LEMMA 3.4

Recall that $S(\mathcal{Z}) = \int_0^1 \mathbb{E} \|Z_1 - Z_0 - v_t(Z_t)\|_2^2 \, dt$. Also, note that $Z_1 - Z_0 = \int_0^1 v_u(Z_u) \, du$. Therefore, we have

$$
\begin{aligned}
S(\mathcal{Z}) &= \int_0^1 \mathbb{E} \left\| \int_0^1 [v_u(Z_u) - v_t(Z_t)] \, du \right\|_2^2 \, dt \\
&= \int_0^1 \mathbb{E} \left\| \int_0^1 \int_t^u \dot{v}_\tau(Z_\tau) \, d\tau \, du \right\|_2^2 \, dt \\
&\leq \int_0^1 \mathbb{E} \left[ \int_0^1 |t - u| \int_{t \wedge u}^{t \vee u} \|\dot{v}_\tau(Z_\tau)\|_2^2 \, d\tau \, du \right] \, dt \\
&\leq \int_0^1 \mathbb{E} \int_0^1 \int_0^1 \|\dot{v}_\tau(Z_\tau)\|_2^2 \, d\tau \, du \\
&\leq \int_0^1 \mathbb{E} \|\dot{v}_\tau(Z_\tau)\|_2^2 \, d\tau = \gamma_1(\mathcal{Z}).
\end{aligned}
$$

Moreover, note that

$$
\gamma_1(\mathcal{Z}) = \sum_{i=1}^T (t_i - t_{i-1}) \cdot \frac{1}{t_i - t_{i-1}} \int_{t_{i-1}}^{t_i} \mathbb{E} \|\dot{v}_\tau(Z_\tau)\|_2^2 \, d\tau \leq \gamma_{2,T}(\mathcal{Z}). \tag{A.9}
$$

This shows the desired inequality.

For the second part, first note that the $t_i - t_{i-1} = 1/T$. Therefore,

$$
\gamma_1(\mathcal{Z}) = \frac{1}{T} \sum_{i=1}^T \frac{1}{t_i - t_{i-1}} \int_{t_{i-1}}^{t_i} \mathbb{E} \|\dot{v}_\tau(Z_\tau)\|_2^2 \, d\tau \geq \frac{\gamma_{2,T}(\mathcal{Z})}{T}.
$$

The above inequality along with (A.9) tells that $\gamma_1(\mathcal{Z}) = 0$ iff $\gamma_{2,T}(\mathcal{Z}) = 0$.

Now, due to the inequality $S(\mathcal{Z}) \leq \gamma_1(\mathcal{Z})$, we have $S(\mathcal{Z}) = 0$ if $\gamma_1(\mathcal{Z}) = 0$. For the other direction, let us assume $S(\mathcal{Z}) = 0$. This shows that $v_t(Z_t) = Z_1 - Z_0$ almost surely in $t$ and $(Z_0, Z_1)$. This shows that $\dot{v}_t(Z_t) = 0$ almost surely. Hence the result follows.

A.2.3  PROOF OF THEOREM 3.5

Recall that for a given partition $0 = t_0 < t_1 < \ldots < t_T = 1$ of the interval $[0, 1]$ of equidistant points $\{t_i\}_{0 \leq i \leq T}$ with $h := T^{-1}$, we follow the Euler discretized version of the of ODE (4) to obtain the sample estimates:

$$
\widehat{Y}_{t_i} = \widehat{Y}_{t_{i-1}} + h \widehat{v}_{t_i}(\widehat{Y}_{t_i}), \quad \widehat{Y}_0 = Z_0.
$$

Before analyzing the discretization error, we introduce the following interpolation process for $t \in [t_i, t_{i+1}]$ and each $i \in \{0, \ldots, T\}$:

$$
\frac{\mathrm{d}}{\mathrm{d}t} \bar{Y}_t = \widehat{v}_{t_i}(\bar{Y}_{t_i}), \quad \bar{Y}_{t_i} = \widehat{Y}_{t_i}. \tag{A.10}
$$

The above ODE flow gives us a continuous interpolation between $\widehat{Y}_{t_i}$ and $\widehat{Y}_{t_{i+1}}$. Coupled with the above flow equation and the ODE flow (5), we have the following *almost sure* differential inequality for $t \in [t_i, t_{i+1}]$:

$$
\begin{aligned}
\frac{\mathrm{d}}{\mathrm{d}t} \|Z_t - \bar{Y}_t\|_2^2 &= 2 \left\langle Z_t - \bar{Y}_t, \frac{\mathrm{d}}{\mathrm{d}t} Z_t - \frac{\mathrm{d}}{\mathrm{d}t} \bar{Y}_t \right\rangle \\
&= 2 \left\langle Z_t - \bar{Y}_t, v_t(Z_t) - \widehat{v}_{t_i}(\bar{Y}_{t_i}) \right\rangle \\
&\leq \widehat{L} \|Z_t - \bar{Y}_t\|_2^2 + \|v_t(Z_t) - \widehat{v}_{t_i}(\bar{Y}_{t_i})\|_2^2 / \widehat{L}
\end{aligned} \tag{A.11}
$$

Multiplying $e^{-\widehat{L}(t-t_i)}$ on both sides of the above inequality and rearranging the terms leads to

$$e^{-\widehat{L}(t-t_i)}\frac{\mathrm{d}}{\mathrm{dt}}\|Z_t - \bar{Y}_t\|_2^2 - e^{-\widehat{L}(t-t_i)}\widehat{L}\|Z_t - \bar{Y}_t\|_2^2 \le e^{-\widehat{L}(t-t_i)}\|v_t(Z_t) - \widehat{v}_{t_i}(\bar{Y}_{t_i})\|_2^2/\widehat{L}.$$

$$\Leftrightarrow \frac{\mathrm{d}}{\mathrm{dt}}\{e^{-\widehat{L}(t-t_i)}\|Z_t - \bar{Y}_t\|_2^2\} \le e^{-\widehat{L}(t-t_i)}\|v_t(Z_t) - \widehat{v}_{t_i}(\bar{Y}_{t_i})\|_2^2/\widehat{L} \le \|v_t(Z_t) - \widehat{v}_{t_i}(\bar{Y}_{t_i})\|_2^2/\widehat{L}.$$

$$\Leftrightarrow \|Z_{t_{i+1}} - \widehat{Y}_{t_{i+1}}\|_2^2 \le e^{\widehat{L}(t_{i+1}-t_i)}\|Z_{t_i} - \widehat{Y}_{t_i}\|_2^2 + \frac{e^{\widehat{L}(t_{i+1}-t_i)}}{\widehat{L}}\int_{t_i}^{t_{i+1}}\|v_t(Z_t) - \widehat{v}_{t_i}(\bar{Y}_{t_i})\|_2^2\,dt.$$

Define $\Delta_i := \mathbb{E}\|Z_{t_i} - \widehat{Y}_{t_i}\|_2^2$. Using the above inequality we have

$$\Delta_{i+1}$$

$$\le e^{\widehat{L}h}\Delta_i + \frac{e^{\widehat{L}h}}{\widehat{L}}\int_{t_i}^{t_{i+1}}\mathbb{E}\|v_t(Z_t) - \widehat{v}_{t_i}(\widehat{Y}_{t_i})\|_2^2\,dt$$

$$\le e^{\widehat{L}h}\Delta_i + \frac{3e^{\widehat{L}h}}{\widehat{L}}\left\{\underbrace{\int_{t_i}^{t_{i+1}}\mathbb{E}\|v_t(Z_t) - v_{t_i}(Z_{t_i})\|_2^2\,dt}_{T_1} + \underbrace{\int_{t_i}^{t_{i+1}}\mathbb{E}\|v_{t_i}(Z_{t_i}) - \widehat{v}_{t_i}(Z_{t_i})\|_2^2\,dt}_{T_2} + \underbrace{\int_{t_i}^{t_{i+1}}\mathbb{E}\|\widehat{v}_{t_i}(Z_{t_i}) - \widehat{v}_{t_i}(\widehat{Y}_{t_i})\|_2^2\,dt}_{T_3}\right\}.$$

$$\text{(A.12)}$$

Now we will bound each of the last three terms on the right-hand side of the above inequality.

**Bounding $T_1$.** For the first term, we have

$$\mathbb{E}\|v_t(Z_t) - v_{t_i}(Z_{t_i})\|_2^2 = \mathbb{E}\left\|\int_{t_i}^t \frac{\mathrm{d}}{\mathrm{d}\tau}v_\tau(Z_\tau)\,d\tau\right\|_2^2$$

$$\le (t-t_i)\int_{t_i}^t \mathbb{E}\left\|\frac{\mathrm{d}}{\mathrm{d}\tau}v_\tau(Z_\tau)\right\|_2^2\,d\tau \qquad\text{(A.13)}$$

$$\le h^2\gamma_i,$$

where $\gamma_i = \frac{1}{t_{i+1}-t_i}\int_{t_i}^{t_{i+1}}\mathbb{E}\|\frac{\mathrm{d}}{\mathrm{d}\tau}v_\tau(Z_\tau)\|_2^2\,d\tau$. This shows that $T_1 \le h^3\gamma_i$.

**Bounding $T_2$.** The term $T_2$ is bounded by $h\varepsilon_{\mathrm{vl}}^2$ as $\mathbb{E}\|v_{t_i}(Z_{t_i}) - \widehat{v}_{t_i}(Z_{t_i})\|_2^2 \le \varepsilon_{\mathrm{vl}}^2$ (Assumption 3.1(a)).

**Bounding $T_3$.** For the final term we will use that $\widehat{v}_{t_i}$ is $\widehat{L}$-Lipschitz. This entails that $T_3 \le \widehat{L}^2 h\Delta_i$. Plugging these bounds in the recursion formula (A.12), we get

$$\Delta_{i+1} \le e^{\widehat{L}h}(1 + 3\widehat{L}h)\Delta_i + 3e^{\widehat{L}h}(h^3\gamma_i + h\varepsilon_{\mathrm{vl}}^2)/\widehat{L}.$$

Solving the recursion yields

$$\Delta_T \le e^{T\widehat{L}h}(1 + 3\widehat{L}h)^T\Delta_0 + \frac{3h^3}{\widehat{L}}\left\{\sum_{k=1}^T e^{k\widehat{L}h}(1 + 3\widehat{L}h)^{k-1}\gamma_{T-k}\right\} + \frac{3h}{\widehat{L}}\left\{\sum_{k=1}^T e^{k\widehat{L}h}(1 + 3\widehat{L}h)^{k-1}\right\}\varepsilon_{\mathrm{vl}}^2.$$

Recall that $\gamma_{2,T}(\mathcal{Z}) := \max_k \gamma_k$. Note that $\Delta_0 = 0$ as $Z_0 = \widehat{Y}_0$. Therefore, we have

$$\Delta_T \le \frac{e^{4\widehat{L}}}{\widehat{L}^2}\left(\frac{\gamma_{2,T}(\mathcal{Z})}{T^2} + \varepsilon_{\mathrm{vl}}^2\right).$$

Here we used the fact that

$$\sum_{k=1}^T e^{k\widehat{L}h}(1 + 3\widehat{L}h)^{k-1} \le \frac{e^{4\widehat{L}} - 1}{1 + 3\widehat{L}h - e^{-\widehat{L}h}} \le \frac{e^{4\widehat{L}}}{3\widehat{L}h}.$$

Therefore, we have

$$W_2^2(\widehat{\rho}_{\mathrm{data}}, \rho_1) \le \Delta_T \le \frac{e^{4\widehat{L}}}{\widehat{L}^2}\left(\frac{\gamma_{2,T}(\mathcal{Z})}{T^2} + \varepsilon_{\mathrm{vl}}^2\right).$$

However, the above upper bound explodes for $\widehat{L} \to 0$. Therefore, we handle the case $\widehat{L} < 1$ in a slightly different manner.

**Separately handling $\widehat{L} < 1$ case:** We recall the decomposition (A.11). We will only change the last inequality in that decomposition, i.e., for $\alpha > 0$ we get

$$
\begin{aligned}
\frac{\mathrm{d}}{\mathrm{dt}} \|Z_t - \bar{Y}_t\|_2^2 &= 2 \left\langle Z_t - \bar{Y}_t, \frac{\mathrm{d}}{\mathrm{dt}} Z_t - \frac{\mathrm{d}}{\mathrm{dt}} \bar{Y}_t \right\rangle \\
&= 2 \left\langle Z_t - \bar{Y}_t, v_t(Z_t) - \widehat{v}_{t_i}(\bar{Y}_{t_i}) \right\rangle \\
&\leq \alpha \|Z_t - \bar{Y}_t\|_2^2 + \|v_t(Z_t) - \widehat{v}_{t_i}(\bar{Y}_{t_i})\|_2^2 / \alpha
\end{aligned}
\tag{A.14}
$$

Therefore, following exactly similar steps as before, we arrive at the following recursion:

$$
\Delta_{i+1} \leq e^{\alpha h} \left( 1 + \frac{3\widehat{L}^2 h}{\alpha} \right) \Delta_i + \frac{3e^{\alpha h}}{\alpha} (h^3 \gamma_i + h \varepsilon_{\mathrm{vl}}^2).
$$

Solving this yields

$$
\Delta_T \leq \frac{e^{\alpha + 3\widehat{L}^2/\alpha} - 1}{1 + 3\widehat{L}^2 h/\alpha - e^{-\alpha h}} \left( \frac{3h^3}{\alpha} \cdot \gamma_{2,T}(\mathcal{Z}) + \frac{3h}{\alpha} \cdot \varepsilon_{\mathrm{vl}}^2 \right)
$$

Note that $e^{\alpha + 3\widehat{L}^2/\alpha} - 1 \leq e^{\alpha + 3\widehat{L}/\alpha} - 1$ as $\widehat{L} < 1$. Additionally,

$$
1 + 3\widehat{L}^2 h/\alpha - e^{-\alpha h} \geq 1 - e^{-\alpha h} \geq \alpha h e^{-\alpha h}.
$$

Setting $\alpha = 1$, and using the above inequalities along with the fact that $h \leq 1$, we get

$$
\frac{e^{\alpha + 3\widehat{L}^2/\alpha} - 1}{1 + 3\widehat{L}^2 h/\alpha - e^{-\alpha h}} \leq \frac{e^{2+4\widehat{L}}}{h}.
$$

Finally, using the above inequality we have

$$
W_2^2(\widehat{\rho}_{\mathrm{data}}, \rho_1) \leq \Delta_T \leq 27 e^{4\widehat{L}} \left( \frac{\gamma_{2,T}(\mathcal{Z})}{T^2} + \varepsilon_{\mathrm{vl}}^2 \right).
$$

Combining this with previous upper bound we finally get the result.

## A.3 Proofs of Section 4

### A.3.1 Proof of Theorem 4.3

If Assumption 4.2 holds, then by Theorem 5.2 of Kunita (1984), we know the solution $Z_t(z_0)$ *exists uniquely* for every $z_0 \in \mathbb{R}^d$. Next, Condition (7) ensures that the ODE is non-explosive (Kunita, 1984, Definiton 5.1, 5.5) within $t \in [0, 1]$. Therefore, by Theorem 5.4 in Kunita (1984), we have $z_0 \mapsto Z_t(z_0)$ to be a $\mathcal{C}^1$ function for all $t \in (0, 1]$, i.e., $J_1^{z_0} := \nabla_{z_0} Z_1(z_0)$ exists. However, $J_1^{z_0}$ might not be invertible. However, this is not a problem as we only need almost sure invertibility of $H_t(Z_0)$ in $(Z_0, t) \sim N(0, I_d) \times \mathrm{Unif}([0, 1])$ which we will show in the subsequent discussion.

Showing straightness of 1-RF is equivalent to showing

$$
\mathbb{E}[Z_1 - Z_0 \mid tZ_1 + (1-t)Z_0] \overset{a.s.}{=} Z_1 - Z_0.
$$

Recall that, showing the above ultimately hinges on showing that the map

$$
H_t(z_0) := tZ_1(z_0) + (1-t)z_0
$$

is a *1-to-1 map* for all $t \in [0, 1]$, where the map $Z_1 : \mathbb{R}^d \to \mathbb{R}^d$ is defined in (6).

We will leverage the well-known Inverse Function Theorem (Hörmander, 2003, Theorem 1.1.7) to show that $H_t$ is a 1-to-1 map. In particular, we will show that for any choice of $z_0 \in \mathbb{R}^d$, the Jacobian $\nabla_{z_0} H_t(z_0) = t\nabla_{z_0} Z_1(z_0) + (1-t)I_d$ has full rank for all $t < 1$. Therefore, $H_t$ is invertible locally around $z_0$.

Let $\Lambda_1^{z_0}$ be the set of complex eigenvalues of $J_1^{z_0}$. The only cases when $\nabla_{z_0} H_t(z_0)$ is singular is when

$$t \in \left\{ \frac{1}{1-\lambda} \mid \lambda \in \Lambda_1^{z_0} \right\} \cap [0,1].$$

Therefore, $\mathbb{P}_{t \sim \text{Unif}([0,1])} \left[ \det(\nabla_{z_0} H_t(z_0)) = 0 \right] = 0$. Now, we need to incorporate randomness in $Z_0$. Note that

$$\mathbb{P}_{(Z_0,t) \sim N(0,I_d) \otimes \text{Unif}([0,1])} \left[ \det(\nabla_{z_0} H_t(z_0)|_{z_0=Z_0}) = 0 \right]$$

$$= \int \mathbb{P} \left( \det(\nabla_{z_0} H_t(Z_0)) = 0 \mid Z_0 = z_0 \right) \rho_0(z_0) \, dz_0$$

$$= \int \mathbb{P} \left( \det(\nabla_{z_0} H_t(z_0)) = 0 \mid Z_0 = z_0 \right) \rho_0(z_0) \, dz_0$$

$$= \int \mathbb{P}_{t \sim \text{Unif}([0,1])} \left( \det(\nabla_{z_0} H_t(z_0)) = 0 \right) \rho_0(z_0) \, dz_0$$

$$= 0.$$

By a similar argument, we can also conclude that

$$0 = \mathbb{P}_{(Z_0,t) \sim N(0,I_d) \otimes \text{Unif}([0,1])} \left[ \det(\nabla_{z_0} H_t(z_0)|_{z_0=Z_0}) = 0 \right]$$

$$= \int \mathbb{P} \left( \det(\nabla_{z_0} H_t(Z_0)) = 0 \mid t = t' \right) \, dt'$$

$$= \int \mathbb{P}_{Z_0 \sim N(0,I_d)} \left( \det(\nabla_{z_0} H_{t'}(Z_0)) = 0 \right) \, dt'.$$

This shows that $\mathcal{S}_t := \{z_0 : \det(\nabla_{z_0} H_t(z_0)) = 0\}$ is a measure-zero set almost everywhere in $t \sim \text{Unif}([0,1])$. Therefore, the set $\mathcal{T} := \{t \in [0,1] : \mathbb{P}_{Z_0 \sim \rho_0}(\mathcal{S}_t) = 0\}$ is an almost sure set in $t \sim \text{Unif}([0,1])$.

**Showing straightness:** For a fix $R > 0$, we have

$$V(Z_0, Z_1) := \mathbb{E}_{Z_0,t} \| Z_1 - Z_0 - \mathbb{E}\{Z_1 - Z_0 \mid H_t(Z_0)\} \|_2$$

$$= \int_0^1 \mathbb{E}_{Z_0} \| Z_1 - Z_0 - \mathbb{E}\{Z_1 - Z_0 \mid H_t(Z_0)\} \|_2 \, dt$$

$$= \int_{\mathcal{T}} \mathbb{E}_{Z_0} \| Z_1 - Z_0 - \mathbb{E}\{Z_1 - Z_0 \mid H_t(Z_0)\} \|_2 \, dt$$

$$= \int_{\mathcal{T}} \underbrace{\mathbb{E}_{Z_0} \left[ \| Z_1 - Z_0 - \mathbb{E}\{Z_1 - Z_0 \mid H_t(Z_0)\} \|_2 \, \mathbb{1}\{\|Z_0\|_2 \leq R\} \right]}_{V_{1,R}(t)} \, dt$$

$$+ \int_{\mathcal{T}} \underbrace{\mathbb{E}_{Z_0} \left[ \| Z_1 - Z_0 - \mathbb{E}\{Z_1 - Z_0 \mid H_t(Z_0)\} \|_2 \, \mathbb{1}\{\|Z_0\|_2 > R\} \right]}_{V_{2,R}(t)} \, dt.$$

**Analyzing $V_{1,R}(t)$:** Note that $\mathcal{S}_t$ is a measure zero set under $\rho_0$. Now, we will construct a finite open cover of $\overline{\mathbb{B}(0,R)}$ in a particular way. First, we focus on $\mathcal{S}_t^c \cap \overline{\mathbb{B}(0,R)}$. Note that for each $z \in \mathcal{S}_t^c \cap \overline{\mathbb{B}(0,R)}$ we can construct an open set $U_{z,t} \ni z$, such that $H_t$ invertible in $U_{z,t}$. Secondly, note that $\mathcal{S}_t \cap \overline{\mathbb{B}(0,R)}$ is a measure-zero set under $\rho_0$. As $\rho_0$ is an *outer regular* measure, for a given $\eta > 0$, we can construct and open set $O_{\eta,t} \supseteq \mathcal{S}_t \cap \overline{\mathbb{B}(0,R)}$ such that $\mathbb{P}(O_{\eta,t}) \leq \mathbb{P}(\mathcal{S}_t \cap \overline{\mathbb{B}(0,R)}) + \eta = \eta$. Therefore,

$$U_t := \left( \cup_{z \in \mathcal{S}_t^c \cap \overline{\mathbb{B}(0,R)}} U_{z,t} \right) \cup O_{\eta,t}$$

is open cover of $\overline{\mathbb{B}(0,R)}$, and we can find a finite cover of $\overline{\mathbb{B}(0,R)}$:

$$U_t^{\text{finite}} := \left( \cup_{i \in [M_t]} U_{z_i,t} \right) \cup O_{\eta,t}$$

Let $\alpha(z_i, t) = \mathbb{P}(Z_0 \in U_{z_i,t} \cap \overline{\mathbb{B}(0, R)})$. Then, we have

$$
\begin{aligned}
V_{1,R}(t) &\leq \mathbb{E}_{Z_0}\big[\|Z_1 - Z_0 - \mathbb{E}\{Z_1 - Z_0 \mid H_t(Z_0)\}\|_2 \, \mathbb{1}\{Z_0 \in U_t^{\text{finite}}\}\big] \\
&\leq \sum_{i \in [M_t]} \alpha(z_i, t) \underbrace{\mathbb{E}_{Z_0 \mid Z_0 \in U_{z_i,t} \cap \overline{\mathbb{B}(0,R)}}\big[\|Z_1 - Z_0 - \mathbb{E}\{Z_1 - Z_0 \mid H_t(Z_0)\}\|_2\big]}_{=0} \\
&\quad + \mathbb{E}_{Z_0}\big[\|Z_1 - Z_0 - \mathbb{E}\{Z_1 - Z_0 \mid H_t(Z_0)\}\|_2 \, \mathbb{1}\{Z_0 \in O_{\eta,t}\}\big]
\end{aligned}
\tag{A.15}
$$

The first term on the right-hand side is 0 *(almost surely)* because of the local invertability of $H_t$ in $U_{z_i,t} \cap \overline{\mathbb{B}(0, R)}$. For the second term, first note that $\mathbb{E}_{Z_0}\big[\|Z_1 - Z_0 - \mathbb{E}\{Z_1 - Z_0 \mid H_t(Z_0)\}\|_2\big] \leq 2(\mathbb{E}\|Z_0\|_2 + \mathbb{E}\|Z_1\|_2) < \infty$. Also, $\lim_{\eta \downarrow 0} \mathbb{P}(O_{\eta,t}) = 0$. Therefore, we have $\lim_{\eta \downarrow 0} \mathbb{E}_{Z_0}\big[\|Z_1 - Z_0 - \mathbb{E}\{Z_1 - Z_0 \mid H_t(Z_0)\}\|_2 \, \mathbb{1}\{Z_0 \in O_{\eta,t}\}\big] = 0$. As the inequality in (A.15) holds for all $\eta > 0$, we have $V_{1,R}(t) = 0$.

**Analyzing $V_{2,R}(t)$:** Lastly, for $V_{2,R}(t)$, we note that by a simple application of dominated convergence theorem (as $\mathbb{E}\|Z_1\|_2 < \infty$) one can conclude $V_{2,R}(t) \to 0$ as $R \uparrow \infty$.

Therefore, taking $R \uparrow \infty$, we can conclude that $V(Z_0, Z_1) = 0$, i.e., $(Z_0, Z_1)$ is a straight coupling.

### A.3.2 Proof of Theorem 4.5

We start by analyzing the velocity function. Recall that

$$
v_t(x) = \begin{cases} \frac{x}{t} + \left(\frac{1-t}{t}\right) s_t(x) & , 0 < t < 1 \\ \mathbb{E}(X_1) - x & , \quad t = 0 \\ x & , \quad t = 1. \end{cases}
$$

where $s_t(x)$ is the (data) score function of $(1-t)X_0 + tX_1$. Let $\phi$ denote the standard gaussian density function in $\mathbb{R}^d$.

**Assumption 4.2:** For $t \in [0, 1)$ we have

$$
\begin{aligned}
s_t(x) &= \nabla_x \log\left(\int_{-\infty}^{\infty} (1-t)^{-d/2} \phi\left(\frac{x - ty}{1-t}\right) \rho_1(dy)\right) \\
&= \frac{\frac{1}{1-t} \int_{-\infty}^{\infty} \left(\frac{ty-x}{1-t}\right) \phi\left(\frac{x-ty}{1-t}\right) \rho_1(dy)}{\int_{-\infty}^{\infty} \phi\left(\frac{x-ty}{1-t}\right) \rho_1(dy)} \\
&= \frac{t}{(1-t)^2} \cdot \frac{\int_{-\infty}^{\infty} y \phi\left(\frac{x-ty}{1-t}\right) \rho_1(dy)}{\int_{-\infty}^{\infty} \phi\left(\frac{x-ty}{1-t}\right) \rho_1(dy)} - \frac{x}{(1-t)^2}.
\end{aligned}
$$

Therefore, $v_t(x) = \frac{\int_{-\infty}^{\infty} \left(\frac{y-x}{1-t}\right) \phi\left(\frac{x-ty}{1-t}\right) \rho_1(dy)}{\int_{-\infty}^{\infty} \phi\left(\frac{x-ty}{1-t}\right) \rho_1(dy)}$ for $t \in [0, 1)$.

It is quite clear that $v_0(x)$ and $v_1(x)$ are $\mathcal{C}^2$ functions. Moreover, one can show that $v_t(x)$ is also $\mathcal{C}^2$ function for every $t \in (0, 1)$ ($\nabla_x$ and $\int$ are interchangeable due to moment condition). It suffices to show that $\Psi_1(x) := \int_{-\infty}^{\infty} y \phi\left(\frac{x-ty}{1-t}\right) \rho_1(dy)$ and $\Psi_2(x) := \int_{-\infty}^{\infty} \phi\left(\frac{x-ty}{1-t}\right) \rho_1(dy)$ are $\mathcal{C}^2$ functions and $\Psi_2 > 0$. Note that, $\Psi_2(x) = \mathbb{E}_{X_1 \sim \rho_1} \phi\left(\frac{x-tX_1}{1-t}\right) > 0$. Now, we will show that $\Psi_1(x)$ is $\mathcal{C}^1$. One can similarly show that it is also $\mathcal{C}^2$ by following a similar argument.

We define

$$
D(x, y) := \nabla_x \left[y \phi\left(\frac{x - ty}{1 - t}\right)\right] = \frac{1}{(1-t)^2} y (ty - x)^\top \exp\left(-\frac{\|x - ty\|_2^2}{2(1-t)^2}\right).
$$

Note that if $\|y\|_2^2 \geq 4\|x\|_2^2/t^2$, we have $\langle u, D(x,y)u \rangle \leq \frac{t\|y\|_2^2 + \|y\|_2 \|x\|_2}{(1-t)^2} \exp(-t\|y\|_2^2/4)$ for all $u \in \mathbb{S}^{d-1}$, as $\|ty - x\|_2^2 \geq (t^2/2)\|y\|_2^2 - \|x\|_2^2 \geq (t^2/4)\|y\|_2^2$. In addition, the upper bound is integrable w.r.t $\rho_1(dy)$.

For $\|y\|_2^2 \leq 4\|x\|_2^2/t^2$, we have $\langle u, D(x,y)u \rangle \leq \frac{t\|y\|_2^2 + \|y\|_2\|x\|_2}{(1-t)^2} \leq \frac{6\|x\|_2^2}{t(1-t)^2}$, and the upper bound is obviously integrabel w.r.t $\rho_1(dy)$. Therefore, we have

$$\nabla \Psi_1(x) = \int_{-\infty}^{\infty} D(x,y) \, \rho_1(dy).$$

The continuity also follows from generalized DCT. One can take a further derivative to show that $\Psi_1$ is $\mathcal{C}^2$ function, and follow the similar argument for $\Psi_2(x)$.

**Non-explosive:** For notational brevity, we write $X_t$ instead of $X_t(z_0)$. Note that

$$\frac{\mathrm{d}}{\mathrm{d}t}\|X_t\|_2^2 = \langle X_t, v_t(X_t) \rangle \leq h(\|X_t\|_2^2).$$

Write $U_t := \|X_t\|_2^2$. Let $V_t$ be a sequence of maps such that

$$\frac{\mathrm{d}}{\mathrm{d}t}V_t = h(V_t); \quad V_0 = U_0.$$

Due to Condition (8), we have $V_t < \infty$. Next, we claim that $U_t \leq V_t$ for all $t \in [0,1]$.

*Under local-lipschitz property:* If not, then there exist times $t_0, t_1$ such that

$$U_{t_0} = V_{t_0}, \quad \text{and} \quad U_t > V_t \quad \text{for all } t_0 < t \leq t_1.$$

Define $\Delta(t) := U_t - V_t$. Therefore, we have $\Delta(t_0) = 0$ and $\Delta(t) > 0$ for all $t \in (t_0, t_1]$. Let $w = U_{t_0} = V_{t_0}$. Due to local-Lipschitz property of $h$, there exists $\delta_w > 0$ and $L_w > 0$ such that

$$|w_1 - w| \vee |w_2 - w| < \delta_w \Rightarrow |h(w_1) - h(w_2)| \leq L_w |w_1 - w_2|.$$

Due to continuity of $U_t$ and $V_t$ at $t = t_0$, there exists $\eta > 0$ such that $t + \eta < t_1$ and for all $\eta' \leq \eta$ we have $|U_{t_0+\eta'} - w| \vee |V_{t_0+\eta'} - w| < \delta_w$. For , $t \in [t_0, t_0 + \eta]$, we consider the ODE

$$\begin{aligned}
\dot{\Delta}(t) &= \dot{U}_t - \dot{V}_t \\
&= h(U_t) - h(V_t) \\
&\leq L_w |U_t - V_t| \quad \text{(local-Lipschitzness)} \\
&= L_w \Delta(t) \quad \text{(as } \Delta(t) > 0\text{)}.
\end{aligned}$$

Therefore, by Gronwall's lemma we have $\Delta(t) \leq \Delta(t_0)\exp(L_w t)$. This implies that $\Delta(t) \leq 0$ for $t \in (t_0, t_0 + \eta]$, which is a contradiction to the fact that $\Delta(t) > 0$ for all $t \in (t_0, t_1]$. Hence, we have $U_t \leq V_t < \infty$ for all $t \in [0,1]$. This establishes the non-explosive property (Condition (7)) of the ODE.

*Under strictly increasing property:* In this case, we will show a stronger result, i.e., $U_t < V_t$ for all $t \in (0,1]$. If not, let $\tau := \inf\{t > 0 : U_t \geq V_t\}$. By definition, we have $\tau > 0$ and $U_\tau \geq V_\tau$. This implies that

$$\int_0^\tau h(U_t) - h(V_t) \, dt \geq 0 \Rightarrow \exists s \in (0, \tau) \text{ such that } h(U_s) \geq h(V_s).$$

Therefore, we have $U_s \geq V_s$, which contradicts the definition of $\tau$. Hence, we have $U_t < V_t$ for all $t \in (0,1]$.

Now the result follows by applying Theorem 4.3.

### A.3.3 1-RF YIELDS STRAIGHT COUPLING: GAUSSIAN TO A GENERAL MIXTURE OF GAUSSIAN

First, for notational brevity, we write $\|u\|_\Sigma = \sqrt{u^\top \Sigma^{-1} u}$ for a positive-definite matrix $\Sigma$. Let $X_0 \sim N(0, I_d)$ and $X_1 \sim \sum_{i=1}^K \pi_i N(\mu_i, \Sigma_i)$. Let $X_t = tX_1 + (1-t)X_0$, then we have

$$v_t(x) = \frac{x}{t} + \frac{1-t}{t}s_t(x) \tag{A.16}$$

where, $s_t(x) = \nabla_x \log p_t(x)$ is given by

$$s_t(x) = \sum_i w_{i,t}(x)\Sigma_{i,t}^{-1}(t\mu_i - x),$$

$\Sigma_{i,t} = (1-t)^2 I_d + t^2 \Sigma_i$ and

$$w_{i,t}(x) = \frac{\pi_i \exp\left(\frac{-\|x - t\mu_i\|_{\Sigma_i}^2}{2}\right)}{\sum_j \pi_j \exp\left(\frac{-\|x - t\mu_j\|_{\Sigma_i}^2}{2}\right)}.$$

Therefore, we have

$$v_t(x) = \sum_i w_{i,t}(x)\left(I_d - (1-t)\Sigma_{i,t}^{-1}\right)\frac{x}{t} + (1-t)\sum_i w_{i,t}(x)\Sigma_{i,t}^{-1}\mu_i$$

Note that, if $\lambda$ is an eigenvalue of $\Sigma_i$, then the corresponding eigenvalue of $\frac{1}{t}(I_d - (1-t)\Sigma_{i,t}^{-1})$ is $\frac{t^2(1+\lambda)-1}{(1-t)^2+t\lambda^2} \le (1+\lambda^{-1})$. Therefore, $\left\|\frac{1}{t}(I_d - (1-t)\Sigma_{i,t}^{-1})\right\|_{op} \le 1 + \left\|\Sigma_i^{-1}\right\|_{op} =: A_i$. Similar argument shows that $\left\|\Sigma_{i,t}^{-1}\right\|_{op} \le A_i$. Therefore, we have

$$\langle x, v_t(x)\rangle \le \underbrace{(\max_i A_i)}_{A} \|x\|_2^2 + \underbrace{(\max_i A_i \|\mu_i\|_2)}_{B} \|x\|_2.$$

Therefore, Assumption 4.4 is satisfied with $h(u) = Au + B\sqrt{u}$ which is strictly monotonic function and $\int_{u_0}^{\infty}(Au + B\sqrt{u})^{-1} du = \infty$ for all $u_0 > 0$. Moreover, we have $\mathbb{E}\|X_1\|_2 < \infty$. Therefore, by Theorem 4.5 we conclude that 1-RF yields a straight coupling.

### A.3.4 PROOF OF THEOREM 4.6

Let $X_0 \sim \mathcal{N}(0, I)$ and $X_1 \sim \mathcal{N}(\mu, \Sigma)$. Let $\Sigma_t = t^2\Sigma + (1-t)^2 I$. Then we have that $X_t \sim \mathcal{N}(t\mu, \Sigma_t)$,. Let the density of $X_t$ be $\xi_t$ and the score $s_t(x) = \nabla_x \log \xi_t(x) = \Sigma_t^{-1}(t\mu - x)$. Therefore, by using (A.24), the drift is given by:

$$v(x,t) = \frac{x}{t} + \frac{1-t}{t}\Sigma_t^{-1}(t\mu - x)$$

$$= (1-t)\Sigma_t^{-1}\mu + \frac{1}{t}\left(I - (1-t)\Sigma_t^{-1}\right)x$$

So the ODE we want to solve is given by:

$$\frac{dZ_t}{dt} - \frac{1}{t}\left(I - (1-t)\Sigma_t^{-1}\right)Z_t = (1-t)\Sigma_t^{-1}\mu \tag{A.17}$$

Now we look at the structure of $I - (1-t)\Sigma_t^{-1}$. Let the eigendecomposition of $\Sigma = U\Lambda U^{\top}$. We will assume $\Sigma$ is full rank. So,

$$I - (1-t)\Sigma_t^{-1} = U\Lambda_t U^{\top}$$

where $\frac{\Lambda_t}{t} = \frac{1}{t}\{I - (1-t)(t^2\Lambda + (1-t)^2 I)^{-1}\}$. This can also be written as:

$$\lambda_{t,i} = \frac{1}{t}\left\{1 - \frac{1-t}{t^2\lambda_i + (1-t)^2}\right\} = \frac{t(1+\lambda_i) - 1}{t^2\lambda_i + (1-t)^2}.$$

Substituting this into Equation (A.17), we have:

$$\frac{dZ_t}{dt} - U\text{diag}(\lambda_{t,1}, \ldots, \lambda_{t,d})U^{\top}Z_t = (1-t)\Sigma_t^{-1}\mu.$$

So, we first get the integrating factors of each eigenvalue.

$$I_i(t) = \frac{1}{\sqrt{(1+\lambda_i)t^2 - 2t + 1}}$$

So we have:

$$U\Lambda_t' U^\top Z_t = U\Lambda_t'' U^\top \mu + constant$$

where $\lambda_{t,i}' = \frac{1}{\sqrt{(1+\lambda_i)t^2 - 2t + 1}}$ and $\lambda_{t,i}'' = \frac{t}{\sqrt{(1+\lambda_i)t^2 - 2t + 1}}$

This yields,

$$\Sigma^{-1/2} Z_1 - Z_0 = \Sigma^{-1/2}\mu$$
$$Z_1 = \Sigma^{1/2} Z_0 + \mu \tag{A.18}$$

### A.3.5 PROOF OF LEMMA 4.8

We recall the ODE $\dot{Z}_t = v_t(Z_t)$ with $Z_0 = z_0$. As $x \mapsto v_t(x)$ is uniformly Lipschitz, there exists a unique solution $\{Z_t\}_{t \in [0,1]}$ such that $Z_0 = z_0$. Moreover, the map $H_t : z_0 \mapsto z_t$ is monotonically increasing. To see this, let us assume $z_0 > \tilde{z}_0$, but $z_t < \tilde{z}_t$. Note that $G(\tau) := H_\tau(z_0) - H_\tau(\tilde{z}_0)$ is continuous in $\tau$. Also, $G(0) > 0$ and $G(t) < 0$. By the intermediate value property, there exists a $t_0 \in [0,1]$ such that $G(t_0) = 0$, i.e., $z_{t_0} = \tilde{z}_{t_0}$. This violates the uniqueness condition of the ODE solution. Hence, $H_t$ is monotonically increasing. By monotonicity, it follows that

$$\mathbb{P}(Z_t \le z_t) = \mathbb{P}(H_t(Z_0) \le H_t(z_0)) = \mathbb{P}(Z_0 \le z_0).$$

This finishes the proof.

### A.3.6 GAUSSIAN TO A MIXTURE OF TWO GAUSSIANS IN $\mathbb{R}^2$

**Proposition A.1.** *Consider $\tilde{X}_0 \sim \mathcal{N}(0, I_2)$ and $\tilde{X}_1 \sim \pi\mathcal{N}(\tilde{\mu}_1, \tilde{\Lambda}) + (1-\pi)\mathcal{N}(\tilde{\mu}_2, \tilde{\Lambda})$ where $\Lambda$ is a PSD diagonal matrix in $\mathbb{R}^2$. Then, $(\tilde{Z}_0, \tilde{Z}_1) = \texttt{Rectify}\left(\tilde{X}_0, \tilde{X}_1\right)$ is a straight coupling.*

*Proof.* Let $P$ be the ortho-normal matrix given by $P = \begin{bmatrix} \frac{\tilde{\mu}_2 - \tilde{\mu}_1}{\|\tilde{\mu}_2 - \tilde{\mu}_1\|} & \frac{R(\tilde{\mu}_2 - \tilde{\mu}_1)}{\|\tilde{\mu}_2 - \tilde{\mu}_1\|} \end{bmatrix}$, where $R = \begin{bmatrix} 0 & -1 \\ 1 & 0 \end{bmatrix}$ is the skew-symmetric matrix for a 90-degree rotation. We rotate our space using the linear transformation $P$ and obtain the random variables $X_0 = P\tilde{X}_0 \sim \mathcal{N}(0, I)$ and $X_1 = P\tilde{X}_1 \sim \pi\mathcal{N}(\mu_1, \Lambda) + (1-\pi)\mathcal{N}(\mu_2, \Lambda)$, where $\mu_i = \begin{bmatrix} x_i & y_i \end{bmatrix}^\top = P\tilde{\mu}_i$, $\Lambda = P\tilde{\Lambda}P^\top$. Also note that by the above construction of the transformation $P$, $x_1 = x_2 := x$. We first show that $(Z_0, Z_1) = \text{Rectify}(X_0, X_1)$ is straight and then argue that an invertible transformation does not hamper straightness.

Let $\Lambda_t = t^2\Lambda + (1-t)^2 I$, then, $X_t \sim \rho_t = \pi\mathcal{N}(t\mu_1, \Lambda_t) + (1-\pi)\mathcal{N}(t\mu_2, \Lambda_t)$ and the score of $\rho_t$, denoted by $s_t$ is:

$$s_t(z_t) = \sum_{i=1}^{2} w_{t,i}(z_t)\Lambda_t^{-1}(t\mu_i - z_t)$$

where the quantity $w_{t,1}(z_t) := \frac{1}{1 + \exp(g_t(z_t))}$, $w_{t,2}(z_t) = 1 - w_{t,1}(z_t)$, and

$$g_t(z) = \log\frac{1-\pi}{\pi} - \frac{1}{2}\left((z - t\mu_2)^T\Lambda_t^{-1}(z - t\mu_2) - (z_t - t\mu_1)^T\Lambda_t^{-1}(z - t\mu_1)\right)$$
$$= \log\frac{1-\pi}{\pi} - \frac{1}{2}\left(t(\mu_1 - \mu_2)^T\Lambda_t^{-1}z + t^2(\mu_2^T\Lambda_t^{-1}\mu_2 - \mu_1^T\Lambda_t^{-1}\mu_1)\right)$$
$$= \log\frac{1-\pi}{\pi} - \frac{1}{2}\left(\frac{t(y_1 - y_2)}{t^2\lambda_2 + (1-t)^2}z + t^2(\mu_2^T\Lambda_t^{-1}\mu_2 - \mu_1^T\Lambda_t^{-1}\mu_1)\right) \tag{A.19}$$

Then, using (A.24), the drift is given by

$$v_t(z_t) = \frac{z_t}{t} + \frac{1-t}{t}s_t(z_t) \tag{A.20}$$

$$= \frac{(I - (1-t)\Lambda_t^{-1})}{t}z_t + (1-t)\Lambda_t^{-1}\sum_{i=1}^{2} w_{i,t}(z_t)\mu_i \tag{A.21}$$

$$= \frac{1}{t}\tilde{\Lambda}_t z_t + (1-t)\Lambda_t^{-1}\begin{pmatrix} x \\ \sum_{i=1}^{2} w_{i,t}(z_t)y_i \end{pmatrix} \tag{A.22}$$

where we define $\tilde{\Lambda}_t := I - (1-t)\Lambda_t^{-1} = I - (1-t)(t^2\Lambda + (1-t)^2 I)^{-1}$.

Now we look at the structure of $\tilde{\Lambda}_t$. We will assume $\Lambda = \mathrm{diag}(\lambda_i)$ is full rank, and $\Lambda_t = \mathrm{diag}(t^2\lambda_i + (1-t)^2)$.

The diagonal elements of $\tilde{\Lambda}_t$

$$\tilde{\lambda}_{t,i} = \frac{1}{t}\left(1 - \frac{1-t}{t^2\lambda_i + (1-t)^2}\right) = \frac{t(1+\lambda_i) - 1}{t^2\lambda_i + (1-t)^2}$$

For now, we have:

$$\frac{dZ_{t,1}}{dt} = \frac{t(1+\lambda_1)-1}{t^2\lambda_1 + (1-t)^2}Z_{t,1} + \frac{1-t}{t^2\lambda_1 + (1-t)^2}x$$

Define the integrating factor $I(t) = \exp\left(-\int_0^t \frac{(1+\lambda_1)u-1}{(1+\lambda_1)u^2 - 2u + 1}\right)\,du = \frac{1}{\sqrt{(1+\lambda_1)t^2 - 2t + 1}}$.

Multiplying $I(t)$ in both sides of the above ODE and integrating in $t \in [0,1]$ we get the following *almost sure* inequality:

$$\frac{Z_{1,1}}{\sqrt{\lambda_1}} - Z_{0,1} = x\int_0^1 \frac{1-t}{((1+\lambda_1)t^2 - 2t + 1)^{3/2}}\,dt = x\left[\frac{t}{\sqrt{(1+\lambda_1)t^2 - 2t + 1}}\right]_0^1 = \frac{1}{\sqrt{\lambda_1}}x.$$

For the second coordinate, we have:

$$f(Z_{t,2}) := \frac{dZ_{t,2}}{dt} = \frac{t(1+\lambda_2)-1}{t^2\lambda_2 + (1-t)^2}Z_{t,2} + \frac{1-t}{t^2\lambda_2 + (1-t)^2}\sum_{i=1}^2 w_{i,t}(Z_{t,2})y_i \qquad \text{(A.23)}$$

We check that $|df(z)/dz|$ is bounded. Using the definition of $g_t$ in Equation (A.19)

$$\left|\frac{d}{dz}f(z)\right| \le \left|\frac{t(1+\lambda_2)-1}{t^2\lambda_2 + (1-t)^2}\right| + |y_1 - y_2|\left|\frac{1-t}{t^2\lambda_2 + (1-t)^2}\frac{d}{dz}(g_t(z))\right|$$

$$\le (1+\lambda_2) + \frac{(1+\lambda_2)^2|y_1 - y_2|^2}{\lambda_2^2}$$

Therefore, $z \mapsto f(z)$ is uniformly Lipschitz, and henceforth, by Lemma A.2 the map $\psi : Z_{0,2} \mapsto Z_{1,2}$ is monotonically increasing.

The above discussion entails that 1-rectified flow essentially sends $Z_0$ through a map $\mathcal{T} : \mathbb{R}^2 \to \mathbb{R}^2$ such that

$$Z_1 = \mathcal{T}(Z_0) = \begin{pmatrix} \sqrt{\lambda_1}Z_{0,1} + x \\ \psi(Z_{0,2}) \end{pmatrix}$$

where $\psi$ is only defined through the ODE (A.23). Therefore, for any $t \in [0,1]$ we have the function

$$h_t(w) := (1-t)w + t\mathcal{T}(w) = \begin{pmatrix} (1-t)w_1 + t(\sqrt{\lambda_1}w_1 + x) \\ (1-t)w_2 + t\psi(w_2) \end{pmatrix}$$

to be an invertible function, which essentially leads to the following relationship between the two $\sigma$-fields of interest:

$$\mathscr{F}(h_t(Z_0)) = \mathscr{F}(Z_0) \quad (\mathscr{F}(X) \text{ denotes the sigma-field generated by } X)$$

for all $t \in [0,1]$. Hence, we finally have

$$\mathbb{E}[\mathcal{T}(Z_0) - Z_0 \mid h_t(Z_0)] = \mathbb{E}[\mathcal{T}(Z_0) - Z_0 \mid Z_0] = \mathcal{T}(Z_0) - Z_0.$$

Now, since $P$ is invertible,

$$\begin{aligned}
v_t(\tilde{Z}_t) &= \mathbb{E}\left[\tilde{X}_1 - \tilde{X}_0 \mid t\tilde{X}_1 + (1-t)\tilde{X}_0 = \tilde{Z}_t\right] \\
&= P^{-1}\mathbb{E}\left[P\tilde{X}_1 - P\tilde{X}_0 \mid t\tilde{X}_1 + (1-t)\tilde{X}_0 = \tilde{Z}_t\right] \\
&= P^{-1}\mathbb{E}\left[P\tilde{X}_1 - P\tilde{X}_0 \mid tP\tilde{X}_1 + (1-t)P\tilde{X}_0 = P\tilde{Z}_t\right] \\
&= P^{-1}\mathbb{E}\left[X_1 - X_0 \mid tX_1 + (1-t)X_0 = Z_t\right] \qquad \because (Z_0, Z_1) \text{ is straight} \\
&= P^{-1}(X_1 - X_0) \\
&= \tilde{X}_1 - \tilde{X}_0
\end{aligned}$$

Hence, $(\tilde{Z}_0, \tilde{Z}_1)$ is also straight. This finishes the proof.

□

**Lemma A.2.** *Consider an ODE of the form*

$$\frac{dx_t}{dt} = c_t\, f_t\,(x_t)$$

*for $t \in [0, 1]$ where $x_t \in \mathbb{R}$ and $c_t > 0$ for all $t \in (0, 1]$.*

(a) *If $\frac{\partial f_t(x)}{\partial x} > 0$, i.e., $f_t(x)$ is an increasing function of $x$, then $x_1$ is a monotonically increasing function of the initial condition $x_0$.*

(b) *If $f_t(x)$ is a uniformly Lipschitz function for all $t \in [0, 1]$, then $x_1$ is a monotonically increasing function of the initial condition $x_0$.*

*Proof.* *Part (a):* Let $x_t^1$ and $x_t^2$ be two solutions to the ODE:

$$\frac{dx_t}{dt} = c_t f_t(x_t), \quad t \in [0, 1],$$

corresponding to the initial conditions $x_0^1$ and $x_0^2$, respectively, with $x_0^1 < x_0^2$. We want to show that $x_1^1 < x_1^2$.

Define the difference between the two solutions:

$$\Delta x_t = x_t^2 - x_t^1.$$

Taking the derivative, we get:

$$\frac{d}{dt}\Delta x_t = \frac{d}{dt}(x_t^2 - x_t^1) = c_t\left(f_t(x_t^2) - f_t(x_t^1)\right).$$

Since $\frac{\partial f_t(x)}{\partial x} > 0$, we have $f_t(x_t^2) > f_t(x_t^1)$ for $x_t^2 > x_t^1$, which implies:

$$\frac{d}{dt}\Delta x_t > 0, \text{ whenever } \Delta x_t > 0.$$

Define $t^* := \inf\left\{t \in (0, 1] : \Delta x_t \le 0\right\}$. Due to inverse map theorem we have $\Delta x_{t^*} \le 0$, and $t^* > 0$ as $\Delta x_0 > 0$. Also, note that

$$\int_0^{t_1} \frac{d}{dt}\Delta x_t\, dt = \Delta x_{t^*} - \Delta x_0 < 0.$$

The above inequality entails that there exists $\tau \in (0, t^*)$ such that $\frac{d}{dt}\Delta x_t < 0$, which implies that

$$f_\tau(x_\tau^2) < f_\tau(x_\tau^1)$$
$$\implies x_\tau^2 < x_\tau^1$$
$$\implies \Delta x_\tau < 0$$

This is again a contradiction to the definition of $t^*$ as $\tau < t^*$. Therefore, we have $\Delta x_t > 0$ for all $t \in [0, 1]$. In particular we have $x_1^2 > x_1^1$.

*Part (b):* If $f_t(x)$ is uniformly Lipschitz, then by Picard-Lindelof theorem, for any tuple $(t_0, x_0)$, there exists only solution $\{x_t\}_{t \in [0,1]}$ passing through $x_0$ at time $t_0$.

Now, following the notation in part (a), let $x_t^1 \le x_t^2$ for some $t$. As $H_t : x_0 \mapsto x_t$ is continuous, so is $G(\tau) := H_\tau(x_0^2) - H_\tau(x_0^1)$. However, $G(0) > 0$ and $G(t) \le 0$. By intermediate value property, there exists $t_0 \in (0, t]$, such that $G(t_0) = 0 \Rightarrow x_{t_0}^2 = x_{t_0}^1$. This contradicts the uniqueness property of the ODE solution. therefore, we have $x_t^2 > x_t^1$ for all $t$. Then the result follows by setting $t = 1$

□

A.3.7   PROOF OF PROPOSITION 4.9

*Proof.* Let $\tilde{X}_0, \tilde{X}_1 \in \mathbb{R}^d$ for $d \geq 2$, where $\tilde{X}_0 \sim \mathcal{N}(0, I)$ and $\tilde{X}_1 \sim \sum_{i=1}^2 \pi_i \mathcal{N}(\tilde{\mu}_i, \sigma^2 I)$ with $\sigma^2 = 1$ (for simplicity). We start with the matrix $\tilde{M} = [\tilde{\mu}_1 \quad \tilde{\mu}_2]$ and perform a QR decomposition: $\tilde{M} = \tilde{Q}\tilde{R}$, where $\tilde{Q} \in \mathbb{R}^{d \times 2}$ is an orthonormal matrix that spans the subspace of $\tilde{\mu}_1$ and $\tilde{\mu}_2$.

Next, we extend $\tilde{Q}$ to a complete orthonormal basis for $\mathbb{R}^d$ using $\tilde{Q}' \in \mathbb{R}^{d \times (d-2)}$, which spans the orthogonal complement of the column space of $\tilde{Q}$. We define $Q = \begin{bmatrix} \tilde{Q} & \tilde{Q}' \end{bmatrix}^\top$. This projection guarantees that:

$$Q\tilde{\mu}_1 = (x_1, y_1, 0, \ldots, 0)^\top, \quad Q\tilde{\mu}_2 = (x_2, y_2, 0, \ldots, 0)^\top$$

i.e., only the first two components are non-zero.

To equalize one of the components, we apply a rotation matrix $R(\theta) \in \mathbb{R}^{d \times d}$, which rotates the first two components while leaving the others unchanged:

$$R(\theta) = \begin{bmatrix} \cos\theta & -\sin\theta & 0 \\ \sin\theta & \cos\theta & 0 \\ 0 & 0 & I_{d-2} \end{bmatrix}$$

We set $\theta$ as:

$$\theta = \tan^{-1}\left(\frac{y_2 - y_1}{x_1 - x_2}\right)$$

This ensures that the second components of $R(\theta)Q\tilde{\mu}_1$ and $R(\theta)Q\tilde{\mu}_2$ are identical.

Finally, we define the overall transformation as $P = R(\theta)Q$. This matrix $P \in \mathbb{R}^{d \times d}$ is orthonormal (and hence, invertible) since it is the product of two orthonormal matrices. The transformation $P$, not only makes the last $d - 1$ coordinates of the means identical but also reduces the effective dimension of the flow to two.

Now, we rotate our space using the linear transformation $P$ and obtain the distributions $X_0 = P\tilde{X}_0 \sim \mathcal{N}(0, I)$ and $X_1 = P\tilde{X}_1 \sim \sum_{i=1}^2 \pi_i \mathcal{N}(\mu_i, \Sigma)$, where $\mu_i = P\tilde{\mu}_i$, $\Sigma = P\tilde{\Sigma}P^\top = I$. Also note that by the above construction of the transformation $P$, $\mu_{1,k} = \mu_{2,k} := c_k$. for all $k \in [d] \setminus \{1\}$. We first show that $(Z_0, Z_1) = \text{Rectify}(X_0, X_1)$ is straight and then argue that an invertible transformation does not hamper straightness.
To proceed, we apply the Rectify procedure on $(X_0, X_1)$ and obtain the following ODE:

$$v_t(Z_t) = \frac{dZ_t}{dt} = \frac{(2t-1)Z_t}{\sigma_t^2} + \frac{1-t}{\sigma_t^2}\sum_{i=1}^2 w_i(Z_t)\mu_i$$

For $k \in [d] \setminus \{1\}$, we have that

$$\frac{dZ_{t,k}}{dt} = \frac{(2t-1)Z_{t,k}}{\sigma_t^2} + c_k$$

Hence, using (A.18) the final mapping is just a translation given by $Z_{1,k} = Z_{0,k} + c_k$. However, for the first co-ordinate, for $g_t(Z_{t,1}) = \log\left(\frac{\pi_2}{\pi_1}\right) - \frac{1}{2\sigma_t^2}\left((Z_{t,1} - t\mu_{2,1})^2 - (Z_{t,1} - t\mu_{1,1})^2\right)$, we have

$$\frac{dZ_{t,1}}{dt} = \frac{(2t-1)Z_{t,1}}{\sigma_t^2} + \frac{1-t}{\sigma_t^2}\left(\frac{\mu_{1,1} + \mu_{2,1}\exp\left(g_t(Z_{t,1})\right)}{1 + \exp\left(g_t(Z_{t,1})\right)}\right)$$

The reasoning used to demonstrate straightness from this point forward is identical to that of Proposition A.1. □

A.3.8   PROOF OF PROPOSITION 4.10

*Proof.* Consider $\boldsymbol{\mu}_{01} = (0, a)^\top, \boldsymbol{\mu}_{02} = (0, -a)^\top$ and $\boldsymbol{\mu}_{11} = (a, a)^\top, \boldsymbol{\mu}_{12} = (a, -a)^\top$ for some $a > 0$. Let

$$X_0 \sim 0.5\mathcal{N}(\boldsymbol{\mu}_{01}, I) + 0.5\mathcal{N}(\boldsymbol{\mu}_{02}, I), \quad X_1 \sim 0.5\mathcal{N}(\boldsymbol{\mu}_{11}, I) + 0.5\mathcal{N}(\boldsymbol{\mu}_{12}, I).$$

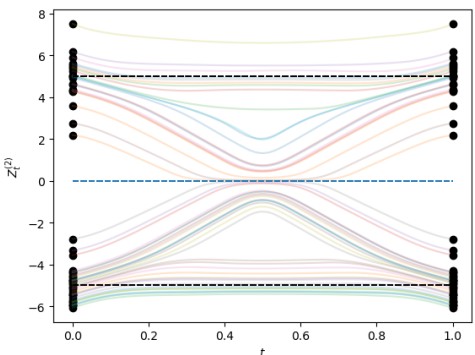

Figure A.4: 1-rectified flow in $y$-direction for GMM

In this case, the velocity functions in $x$ and $y$-direction for 1-rectification turns out to be

$$u_t(x) = \frac{(2t-1)\,x}{\sigma_t^2} + \frac{(1-t)a}{\sigma_t^2},$$

$$v_t(y) = \frac{(2t-1)\,y}{\sigma_t^2}$$

$$+ \frac{a}{\sigma_t^2} \cdot \frac{\exp\left(-\frac{(y-a)^2}{2\sigma_t^2}\right)(1-2t) - \exp\left(-\frac{(y+a)^2}{2\sigma_t^2}\right)(1-2t) + \exp\left(-\frac{(y-(2t-1)a)^2}{2\sigma_t^2}\right) - \exp\left(-\frac{(y+(2t-1)a)^2}{2\sigma_t^2}\right)}{\exp\left(-\frac{(y-a)^2}{2\sigma_t^2}\right) + \exp\left(-\frac{(y+a)^2}{2\sigma_t^2}\right) + \exp\left(-\frac{(y-(2t-1)a)^2}{2\sigma_t^2}\right) + \exp\left(-\frac{(y+(2t-1)a)^2}{2\sigma_t^2}\right)}.$$

Next, we will take the derivative of $v_t(y)$ with respect to $y$. For notational brevity, let us define

$$e_1(y) = \exp\left(-\frac{(y-a)^2}{2\sigma_t^2}\right)(1-2t),$$

$$e_2(y) = \exp\left(-\frac{(y+a)^2}{2\sigma_t^2}\right)(1-2t),$$

$$e_3(y) = \exp\left(-\frac{(y-a(2t-1))^2}{2\sigma_t^2}\right),$$

$$e_4(y) = \exp\left(-\frac{(y+a(2t-1))^2}{2\sigma_t^2}\right).$$

Then we have

$$\left|\frac{dv_t(y)}{dy}\right| \le \frac{2t-1}{\sigma_t^2} + \frac{a^2}{\sigma_t^4} \cdot \frac{4\{e_1(y)e_2(y) + e_2(y)e_3(y) + e_3(y)e_4(y) + e_4(y)e_1(y)\}}{(\sum_{j=1}^4 e_j(y))^2} \le 2 + 4a^2.$$

We used the basic inequalities $4(ab + bc + cd + da) \le (a + b + c + d)^2$ and $\sigma_t^2 \ge 1/2$ in the last step of the above display.

This shows that $v_t(y)$ is uniformly Lipschitz. This entails that the map $\mathcal{T}: \mathbb{R} \to \mathbb{R}$ that sends $y_0$ to a point $y_1 \in \mathbb{R}$, and defined through the ODE

$$\frac{d}{dt}Y_t = v_t(Y_t);\ Y_0 = y_0,$$

is an injective map due to the uniqueness of the solution of the above ODE. Also, we denote by $Y_t^{y_0}$ the solution of the above ODE.

To show the strict increasing property of $\mathcal{T}$, let us consider the same ODE with $Y_0 = \tilde{y}_0 < y_0$. We also consider the solution $Y_t^{\tilde{y}_0}$. Consider the function $L_t := Y_t^{y_0} - Y_t^{\tilde{y}_0}$, which is also continuous in $t \in [0,1]$. To prove

increasing property, it is enough to show that $L_1 > 0$. Let us assume that $L_1 \leq 0$. We already know $L_0 > 0$, and hence by Intermediate Value Property, we have there exists a $\tau \in (0, 1]$ such that $L_\tau = 0$. This entails that there exists $y_\tau \in \mathbb{R}$ such that $Y_\tau^{y_0} = Y_\tau^{\tilde{y}_0} = y_\tau$. This shows that we have two different solutions of the ODE passing through $(\tau, y_\tau)$, which is a contradiction. This proves the coveted strict increasing property of $\mathcal{T}$. Hence, *we have a straight coupling* by similar argument as in previous section. $\square$

### A.4 AUXILIARY RESULTS

#### A.4.1 CONNECTION BETWEEN SCORE AND DRIFT

Let $X_0 = Z \sim \mathcal{N}(0, I)$ and $X_1 = X \sim \rho_{\text{data}}$. Let the density of $X_t = tX + (1-t)Z$ be $\xi_t$. Then Tweedie's formula (Robbins (1992)) gives that $\mathbb{E}\left[tX \mid X_t = x\right] = x + (1-t)^2 s_t(x)$ where $s_t(x) = \nabla \log \xi_t(x)$

We have that

$$
\begin{aligned}
v_t(x) &= \mathbb{E}[X - Z \mid X_t = x] \\
&= \mathbb{E}[\frac{X - X_t}{1 - t} \mid X_t = x] \\
&= \frac{x + (1-t)^2 s_t(x)}{t(1-t)} - \frac{x}{(1-t)} \quad \text{(applying Tweedie's formula)} \\
&= \frac{x}{t} + \left(\frac{1-t}{t}\right) s_t(x)
\end{aligned}
\tag{A.24}
$$

#### A.4.2 AUXILIARY RESULTS FOR GAUSSIAN MIXTURE TO GAUSSIAN MIXTURE FLOW

In this section, we will procure a formula of the drift function for 1-rectified flow from a Gaussian mixture to another Gaussian mixture. Let $X_0 \sim \frac{1}{K_0} \sum_{i=1}^{K_0} \mathcal{N}(\mu_{0i}, \sigma^2 I)$, $X_1 \sim \frac{1}{K_1} \sum_{i=1}^{K_1} \mathcal{N}(\mu_{1i}, \sigma^2 I)$, and $X_t = tX_1 + (1-t)X_0$.

$$
\begin{aligned}
v_t(x) &= \mathbb{E}\left[X_1 - X_0 \mid X_t = x\right] \\
&= \mathbb{E}\left[\frac{X_1 - X_t}{1 - t} \mid X_t = x\right] \\
&= \frac{1}{t(1-t)}\left(\mathbb{E}\left[tX_1 \mid X_t = x\right] - tx\right) \\
&= \frac{1}{t(1-t)}\left(\frac{1}{K_0}\sum_{i=1}^{K_0}\frac{p_t^{(i)}(x)}{p_t(x)}\mathbb{E}\left[tX_1 \mid X_t^{(i)} = x\right] - tx\right) \\
&= \frac{1}{t(1-t)}\left(\frac{1}{K_0}\sum_{i=1}^{K_0}\frac{p_t^{(i)}(x)}{p_t(x)}\left(x - (1-t)\mu_{0i} + \tilde{\sigma}_t^2 s_t^{(i)}(x)\right) - tx\right), \quad \text{where } \tilde{\sigma}_t^2 = (1-t)^2\sigma^2 \\
&= \frac{x}{t} + \frac{(1-t)\sigma^2}{t}\left(\frac{1}{K_0}\sum_{i=1}^{K_0}\frac{p_t^{(i)}(x)}{p_t(x)}\left(s_t^{(i)}(x) - \frac{\mu_{0i}}{1-t}\right)\right)
\end{aligned}
$$

where $p_t^{(i)}(x) = \text{Law}(tX_1 + (1-t)\mathcal{N}(\mu_{0i}, \sigma^2)) = \frac{1}{K_1}\sum_{j=1}^{K_1}\mathcal{N}(\underbrace{t\mu_{1j} + (1-t)\mu_{0i}}_{\mu_{tj}^{(i)}}, \sigma_t^2)$, $\sigma_t^2 = (t^2 + (1-t)^2)\sigma^2$.

$$
s_t^{(i)}(x) = \nabla_x \log p_t^{(i)}(x) = \frac{1}{\sigma_t^2}\left(\sum_{j=1}^{K_1} w_j^{(i)}(x)\mu_{tj}^{(i)} - x\right),
$$

where

$$w_j^{(i)}(x) = \frac{\exp\left(\frac{-\left\|x-\mu_{tj}^{(i)}\right\|^2}{2\sigma_t^2}\right)}{\sum_j \exp\left(\frac{-\left\|x-\mu_{tj}^{(i)}\right\|^2}{2\sigma_t^2}\right)}$$

### A.4.3    GAUSSIAN TO A MIXTURE OF GAUSSIAN CASE

Let $X_0 \sim \mathcal{N}(0, I)$ and $X_1 \sim \sum_i \pi_i \mathcal{N}(\mu_i, \sigma_i^2 I)$. Let $X_t = tX_1 + (1-t)X_0$, then using (A.24), we have

$$v_t(x) = \frac{x}{t} + \frac{1-t}{t} s_t(x) \tag{A.25}$$

where, $s_t(x) = \nabla_x \log p_t(x)$ is given by

$$s_t(x) = \sum_i w_{i,t}(x)\left(\frac{t\mu_i - x}{\sigma_{i,t}^2}\right),$$

$\sigma_{i,t}^2 = (1-t)^2 + t^2\sigma_i^2$ and

$$w_{i,t}(x) = \frac{\pi_i \exp\left(\frac{-\|x-t\mu_i\|^2}{2\sigma_{i,t}^2}\right)}{\sum_j \pi_j \exp\left(\frac{-\|x-t\mu_j\|^2}{2\sigma_{i,t}^2}\right)}$$