# OpenReview forum: "2-Rectifications are Enough for Straight Flows: A Theoretical Insight into Wasserstein Convergence"
_ICLR.cc/2025/Conference — ICLR 2025 Conference Withdrawn Submission_

### Official Review · Reviewer_m3Xn · 2024-11-02

**Soundness:** 3
**Presentation:** 3
**Contribution:** 3
**Rating:** 6
**Confidence:** 4

**Summary:**

The paper provides theoretical analysis on the distribution generated by rectified flow and elaborate on the case of Gaussian mixtures.

**Strengths:**

1. The paper is clearly written and well-organized.
2. The theoretical framework is both novel and broadly applicable to the study of rectified flow, particularly in demonstrating the relationship between RF’s performance, the straightness of the flow, and the discretization accuracy.
3. The authors present experiments on both synthetic and real datasets. The figures, particularly those illustrating the sampling path and straightness, effectively convey the key findings.

**Weaknesses:**

1. Assumption 3.1 regarding the estimation error and the Lipschitz property of the drift function, while common in previous diffusion model research, may not be essential for distribution estimation. These assumptions impose additional requirements on the RF sampling path. For the estimation error assumption, could the authors prove rather than assume a small estimation error when learning the drift function of a target distribution with some regularity assumptions using neural networks? Additionally, could the Lipschitz condition be derived using assumptions based solely on the target distribution rather than on the entire sample path?
2. The tightness and effectiveness of the theoretical results on the 2-Wasserstein bound (Theorems 3.2 and 3.5) are uncertain, as the error is exponentially dependent on the Lipschitz constant $\hat{L}$. Moreover, in theorem 3.5, the error bound even tends to infinity when $\hat{L} \rightarrow 0$, which seems counterintuitive. It would be helpful if the authors could conduct additional experiments to perform a quantitative sensitivity analysis on both $\hat{L}$ and $T$ to demonstrate the tightness of the results.
3. Minor typos:
(Line 220) "on-side" Lipschitzness ->one-side Lipschitzness
(Line 323) Under "he" condition -> Under the condition

**Questions:**

1. Regarding the definition of straightness in Definition 2.1, why straigtness is equivalent to the latter expectation form?
2. Regarding the straightness of RF for Gaussian mixtures, is it possible to extend the theory to multivariate Gaussian mixtures with multiple modes? (more than 2 modes). What are the difficulty for proving the straightness?

---

> ### Author Response · Authors · 2024-11-18
>
> Thank you for your questions and constructive comments. We address your concerns point by point below.
>
> **(P1) Assumption 3.1 regarding the estimation error and the Lipschitz property of the drift function, while common in previous diffusion model research, may not be essential for distribution estimation. These assumptions impose additional requirements on the RF sampling path. For the estimation error assumption, could the authors prove rather than assume a small estimation error when learning the drift function of a target distribution with some regularity assumptions using neural networks? Additionally, could the Lipschitz condition be derived using assumptions based solely on the target distribution rather than on the entire sample path?** -- Thank you for your constructive comments and the interesting questions. We think , one can prove results similar to estimation error bound by imposing assumptions on the neural network. In particular, one can also obtain sample complexity results (under finite VC dimension or similar assumptions) that will guarantee good estimation error. However, in this paper we solely wanted to investigate the efficacy of Rectified flow. However, we do agree that a theory for estimation error is pertinent in such studies and defer it to future research.
>
> We think that it is possible to impose a condition on the target distribution so that the velocity function is Lipschitz. For example, if the pdf $\xi_1$ of the target distribution is fairly smooth with bounded derivatives, then it can be shown that the resulting velocity is Lipschitz. However, one can ensure the lipschitzness of the estimated velocity $\hat v$ by restricting the class of neural networks with bounded weights.
>
> **(P2) The tightness and effectiveness of the theoretical results on the 2-Wasserstein bound (Theorems 3.2 and 3.5 ) are uncertain, as the error is exponentially dependent on the Lipschitz constant $\hat L$. Moreover, in theorem 3.5 , the error bound even tends to infinity when $\hat{L} \rightarrow 0$, which seems counterintuitive. It would be helpful if the authors could conduct additional experiments to perform a quantitative sensitivity analysis on both $\hat{L}$ and $T$ to demonstrate the tightness of the results.**-- We do agree that the upper bound result has room for improvement and it may not be tight. However, we want to point out that this the very first result that connects $W_2$ distance with the straightness of rectified flow. We mainly wanted to highlight the effect of straightness on the generation quality of rectified flow. We will also add some numerical plots to highlight the effect of straightness in the revised version.
>
>  Also, thank you for pointing out the issue regarding $\hat L \to 0$. A refined analysis shows that the bound can be improved to
>  $$
>     W_2^2(\hat \rho_{data}, \rho_1) \lesssim \frac{e^{4 \hat L}}{\hat L^2 \vee 1} \left( \gamma_{2,T}/ T^2 + \varepsilon^2\right).
>  $$
>  The refined bound now circumvents the aforementioned issue. Please refer to the updated Theorem 3.5.
>
> We also provide the plot of $\log(W_2)^2$ vs $L$ [in this link](https://ibb.co/gMCz6JN). It shows that $W_2$ is increasing with $L$. However, the plot also suggests that there is room for improvement in the upper bound in Theorem 3.5. We defer this to future work as mentioned in Section 6.
>
> **(P3) Regarding the definition of straightness in Definition 2.1, why straightness is equivalent to the latter expectation form?**-- Straight flow implies that the velocity $\mathbb{E}(X_1 - X_0 \mid X_t)$ along the path is not changing with time $t$. Let's say it is a constant $c$ independent of $t$. $X_1$ and $X_0$ are the two end-points then we need $X1 = X_0 + c$. In this case velocity is $\mathbb{E}(X1- X_0 \mid X_t)$, and that is why it follows that $\mathbb{E}(X_1 - X_0 \mid X_t) = c = X_1 - X_0$ almost surely. Moreover, if  $\mathbb{E}(X_1 - X_0 \mid X_t) = c = X_1 - X_0$, then velocity is $t$ independent, and hence the trajectory is straight. Also, we rectified a minor error regarding the non-intersecting trajectories in Definition 2.1. Please refer to the updated version.

---

> ### Author Response · Authors · 2024-11-18
>
> **(P4) Regarding the straightness of RF for Gaussian mixtures, is it possible to extend the theory to multivariate Gaussian mixtures with multiple modes? (more than 2 modes). What are the difficulty for proving the straightness?**-- We now have a fairly general result for multivariate distribution, and we have added it as Theorem 4.5 (or Theorem 4.3). Note that, in the new result, target distribution does not need to have a pdf, it only needs to have a finite 1st moment. Also, the non-explosive condition is fairly mild and is enjoyed by a large class of distributions including a general mixture of Gaussians with multiple modes. See discussion in Lines 381-384 and Lines 388-393.
> The main difficulty in the proof arose in showing the invertibility of a certain multivariate map, for which we used the Inverse function theorem. Also, we had to take care of the existence and uniqueness of the ODE under a more general setting.

---

> > ### Comment · Reviewer_m3Xn · 2024-11-18
> > **Response**
> >
> > Thank you for your response! I feel that my quesions have been well addresed. I will keep my positive score.

---

> > > ### Author Response · Authors · 2024-11-27
> > >
> > > We are glad that you found our response helpful. Thank you for handling our submission.

---

### Official Review · Reviewer_ptxD · 2024-11-02

**Soundness:** 2
**Presentation:** 3
**Contribution:** 3
**Rating:** 8
**Confidence:** 3

**Summary:**

In the context of flow-based generative models, the authors provide a theoretical analysis of Rectified Flow, a generative model introduced by Liu et al (2023). Since the generative model has an associated ODE that has to be discretized to be solved in practice, how straight the solution paths of the ODE is a relevant question, where straighter paths can be solved with fewer discretization steps. Rectified Flow provides an iterative method that successively straigthens the paths. The authors provide a theoretical analysis showing that for generating Gaussian Mixtures, two iterations are enough to get straight paths. The authors also analyze the Wasserstein distance between the target distribution and the generated distribution after one step of Rectified Flow, showing that it can be upper bounded in terms of a measure of straightness.

**Strengths:**

* The authors do a good job introducing the background and putting their results into context with what was previously proved, with a clear presentation for their results.
* The 2-Wasserstein bound provided in Theorem 3.5 and Corollary 3.6 are useful result since they gives an explicit dependence on the straightness parameters they defined and furthermore provides a guide on how to discretize to achieve a certain error if we have access to these straightness parameters. The intuition for these parameters and comparison with another definition of straightness parameter is also helpful.
* Theorem 4.4 is a clean and simple illustration of the effect rectifying flows and provides a stepping stone towards understanding rectifying flows in more general data distributions.

Overall, the authors provide an useful theoretical foundations both for understanding the exact effect of rectifying flows when the data is coming from the Gaussian Mixture, and also in bounding $W_2$ between target and data coming from the rectified flow. The new definitions of straightness introduced seem natural and could prove to be useful in future research.

**Weaknesses:**

It is stated and proved that 1-Rectified flows are straight for Gaussian Mixtures. It seems to me that a very important question is not considered: whether there is any example of initial laws for $X_0$ and $X_1$ whose 1-Rectified flow is not straight. Equivalently, we can ask whether there is any example of laws for $X_0$ and $X_1$ whose 1-Rectified coupling is different from the 2-Rectified coupling. I may be missing something, but I tried to find for these counterexamples and did not succeed. I will raise my score if this question is addressed.

It is clear that 1-Rectified flows are always straight in 1 dimension, since any coupling coming from an ODE is monotonically increasing and deterministic so that by Lemma 4.2 is straight. However, say in Proposition 4.5, the intuitive argument given is that the flows along each coordinate decouples. Hence it seems to me that a much simpler proof may just be to show that the coordinates decouple and then just use Lemma 4.2. A similar reasoning may apply to the proof of Theorem 4.4.

The experiments do not seem to illustrate the upper bound on the 2-Wasserstein distance much. Indeed, the fact that the $W_2$ decays as $T$ grows is to be expected even if we did not know about the bound. It would be interesting if we saw in the experiments a clear evidence that $W_2$ decays with $\gamma_{2,T}.$ However, we see in Figure (3)(b) that $\gamma_{2,T}$ for the 2 component Gaussian Mixture is almost identical to the $\gamma_{2,T}$ for the 3 component, but the $W_2$ in Figure (3)(a) is decreasing much faster for the 2 component than for the 3 component. Of course, this does not contradict the bound since it is just an upper bound, but it is not illustrating it well either.

The plots in Figure (3)(a) and (c) are also not very informative since they just show zero for all $T\geq 5.$ It may be better to plot them in log scale. In particular, the bound suggests that there should be a $1/T^2$ decay for the $W_2,$ but since it is not in log scale it is quite hard to tell whether this is True in the empirical data or not.

Smaller comments:
* In between line 216 and 217 it is said that the bound on the 2-Wasserstein is $O(\epsilon^2),$ but that is the bound on the 2-Wasserstein squared.
* The plots on Figure 1 are not displayed correctly, the x label is not showing.
* In line 206 it says $\hat \rho _\text{data}$ it should say $\rho$
* In line 837, there is an incomplete line that is likely missing an equation.
* The proof sketch for Theorem 4.4 is a bit confusing. I.e., the presentation could be improved e.g. it says '(WLOG call this coordiante 1).
 Denote this coordinate by $c_1,$' and 'isoperimetric mixture' without defining it.

**Questions:**

* In between liens 217 and 218, it is said that the requirement on $b(t)$ of $\sup_{t\in[0,1]}b(t)\leq \epsilon^2$ is much more stringent than that of Assumption 3.1(a). Is there actually an application/relevant setting where Assumption 3.1(a) is fulfilled but $\sup_{t\in[0,1]}b(t)\leq \epsilon^2$ is not?
* The points chosen for the experiment on the Gaussian Mixture mentioned in Example 1 are a bit unmotivated to me. Why did you chose them this way?
* See the first paragraph of weaknesses.

---

> ### Author Response · Authors · 2024-11-18
>
> Thank you for your questions and constructive comments. We address your concerns point by point below.
>
>  **(P1) It is stated and proved that 1-Rectified flows are straight for Gaussian Mixtures. It seems to me that a very important question is not considered: whether there is any example of initial laws $X_0$ for $X_1$ and
>  whose 1-Rectified flow is not straight. Equivalently, we can ask whether there is any example of laws for $X_0$
>  and $X_1$ whose 1-Rectified coupling is different...** -- Thank you for this thought-provoking question. Just to be, clear, we show that for the mixture of Gaussian, 1-rectification generates a straight coupling, or equivalently 2-rectification is a straight flow. This is also equivalent to saying that 1-rectified coupling is the same as the 2-rectified coupling as you have already mentioned.
>     We do agree that it is important to understand when straightness is violated. First, we would like to draw your attention to a fairly general result (Theorem 4.5 and Theorem 4.3) that we recently proved. An informal version is displayed below:
>
> *_Theorem 4.5 (informal version)_: Assume $X_0 \sim N(0, I_d)$ and the target data $X_1 \sim \rho_1$ (independent of $X_0$) with finite 1st moment, i.e., $\mathbb{E} \Vert X_1\Vert_2< \infty$. Also, assume that the ODE (3) is non-explosive (essentially solutions $Z_t$ do not diverge to infinity). Then, the resulting rectified coupling $(Z_0,Z_1):= \texttt{Rect}(X_0, X_1)$ under the aforementioned conditions is  a straight coupling*
>
>  Note that, in the theorem, target distribution does not need to have a pdf, it only needs to have a finite 1st moment. Also, the non-explosive condition is fairly mild and is enjoyed by a large class of distributions including a general mixture of Gaussians. Therefore, to come up with a counter-example, we have to look outside the aforementioned examples.
>
> We first want to draw your attention to the following example:
>  $X_0\sim \delta_0$ and $X_1 \sim 0.5 \delta_1 + 0.5 \delta_2$, where $\delta_n$ denotes Dirac measure on $n$. In this case, there is no straight coupling. To see this, assume there is a straight coupling. Then it can be represented by a deterministic map $\psi$ such that $X_1 \overset{d}{=} \psi(X_0)$. This is contradiction as $\psi(X_0) \sim \delta_{\psi(0)}$. In fact, $(X_0, X_1)$ can not be rectified, as rectification always leads to a deterministic coupling which again raises contradiction by the previous logic. However, under the knowledge of the current general theorem,  we suspect that if $(X_0, X_1)$ is rectifiable for some initial choices of distributions, then the 1-rectified flow will result in a straight coupling under very mild regularity conditions. In fact, we conjecture that 1-rectified flow (under certain regularity conditions) will result in the optimal coupling induced by the Monge map (if it exists), as it is known that RF iteratively solves the optimal transport problem [1]. Although, this is an interesting research direction we defer it to future research.
>
>  **(P2) It is clear that 1-Rectified flows are always straight in 1 dimension since any coupling coming from an ODE is monotonically increasing and deterministic so that by Lemma 4.2 is straight. However, say in Proposition 4.5, the intuitive argument given is that the flows along each coordinate decouples. Hence it seems to me that a much simpler proof may just be to show that the coordinates decouple and then just use Lemma 4.2. A similar reasoning may apply to the proof of Theorem 4.4.**-- We would like to point out that all 1-rectified flows in 1-dimension are not straight as shown in the previous example. If the velocity $v(x,t)$ is Lipschitz, then one can argue that the resulting coupling is monotonic as the solutions to the ODE is unique due to Picard-Lideloff theorem.
>
> Next, we would like to point out that the decoupling of the coordinates phenomenon is not a general phenomenon. For example, even in Theorem 4.4, the coordinates do not decouple if the target distribution is $\pi_1 N(\mu_1, \Sigma_1) + \pi_2 N(\mu_2, \Sigma_2)$ for some general choices of $\{\pi_i, \mu_i, \Sigma_i\}_{i \in [2]}$. In the current paper, $\Sigma_i = I_d$ helps us to decouple the co-ordinates. Also, for Proposition 4.4, symmetric choices of the means are crucial for the decoupling argument which may not hold for general choices of the mean vectors.
>
> To this end, we want to Theorem 4.5 which we mentioned in the previous point. Note that, target distribution does not need to have a pdf, it only needs to have a finite 1st moment. Also, the non-explosive condition is fairly mild and is enjoyed by a large class of distributions including a general mixture of Gaussians. Therefore, the aforementioned theorem covers a large class of rectified flows.
>
>
>
>
>
>
>
> [1] Liu, Q. (2022). Rectified flow: A marginal preserving approach to optimal transport. arXiv preprint arXiv:2209.14577.

---

> > ### Author Response · Authors · 2024-11-18
> >
> > **(P3) The experiments do not seem to illustrate the upper bound on the 2-Wasserstein distance much. Indeed, the fact that the $W_2$ decays as $T$ grows is to be expected even if we did not know about the bound. It would be interesting if we saw in the experiments a clear evidence that $W_2$ decays with $\gamma_{2, T}$. However, we see in Figure (3)(b) that $\gamma_{2, T}$ for the 2 component Gaussian Mixture is almost identical to the $\gamma_{2, T}$ for the 3 component, but the $W_2$ in Figure (3)(a) is decreasing much faster for the 2 component than for the 3 component. Of course, this does not contradict the bound since it is just an upper bound, but it is not illustrating it well either.**-- We appreciate the reviewer's constructive feedback here. We acknowledge that the plots in the current version are not very elusive of the dependence of the upper bound on the straightness parameter, and this is owed to a rather random choice of the means of the mixture components. The norm of the means and separation between the means of the components heavily affect the Lipschitz constant of the drift function which in turn directly impacts the Wasserstein bound as stated in Theorem 3.5 in the main paper. This is the exact reason why even though the straightness parameters for 2 and 3 component mixtures are similar, the Wasserstein error differs largely in the current plots. To isolate the direct effect of the Lipschitz constant on the bound and to elucidate the dependence between $\gamma_{2, T}$ and $W_2^2$ more explicitly, we redo the experiment and choose all the means to have the same norm (say $5$) and with similar separation. It is clear in the new plots that since $\gamma_{2, T}$ parameters are close for 3 and 4 component mixtures, the Wasserstein errors are also close. We give relevant plots in the log-log scale as pointed out by the reviewer. We also show that after obtaining a straight coupling from the first rectification, the second rectified flow is much straighter and has an almost zero Wasserstein error even with just one discretized step. In the revised version we will add all these suggested plots.
> >
> > **(P4) The plots in Figure (3)(a) and (c) are also not very informative since they just show zero for all $T \geq 5$. It may be better to plot them in log scale. In particular, the bound suggests that there should be a $1 / T^2$ decay for the $W_2$, but since it is not in $\log$ scale it is quite hard to tell whether this is True in the empirical data or not.**-- In the updated paper, We give all the plots in the log-log scale and show that even on a larger spectrum of discretization steps. The plots suggest that the $1/T^2$ dependence we derive in our theoretical analysis is somewhat tight (before it stabilizes). See discussion in Lines 464-465.
> >
> > **(P5) In between liens 217 and 218 , it is said that the requirement on $b(t)$ of $\sup _{t \in[0,1]} b(t) \leq \epsilon^2$ is much more stringent than that of Assumption 3.1 (a). Is there actually an application/relevant setting where Assumption 3.1(a) is fulfilled but $\sup _{t \in[0,1]} b(t) \leq \epsilon^2$ is not?**-- Thank you for your question. Assumption 3.1(a) is actually more realistic compared to its stronger version. Note that, during real-world training, we have to discretize the time interval $[0,1]$ into the partition $t_j$ for $j \in \{0,.., T\}$. Therefore, during the training phase, the model only gets samples from $\{Law(X_{t_j})\}$ for $j \in \{0,..., T\}$. Therefore, we can only expect small training error only at the points in those $t_j$'s. As the model does not see any data for other $t \in [0,1]$, once can not train the model for those unseen points $t$, and naturally training error is expected to be large, i.e., $\sup_{t\in [0,1]} b(t) \le \epsilon^2$ is indeed unrealistic.
> >
> > **(P6) The points chosen for the experiment on the Gaussian Mixture mentioned in Example 1 are a bit unmotivated to me. Why did you chose them this way?**-- We thank the reviewer for pointing this out and acknowledge that the points were chosen in a random manner. We only wanted the means to be well separated. But as we pointed out in the previous comments, we have redone this experiment with a rather informed choice of the means. For completeness, we reiterate the motivation behind the choice of new means. The norm of the means and separation between the means of the components heavily affect the Lipschitz constant of the drift function, which  directly impacts the Wasserstein error as stated in Theorem 3.5 in the main paper. This led to confusing plots in the current version. To isolate the direct effect of the Lipschitz constant on the bound and to elucidate the dependence between $\gamma_{2, T}$ and $W_2^2$ more explicitly, we redo the experiment and choose all the means to have the same norm (say $5$) and with similar separation.

---

> ### Author Response · Authors · 2024-11-30
> **Request for feedback**
>
> Dear Reviewer ptxD,
>
> Thank you so much for your thoughtful remarks and thorough review. We have added a new general straightness result (Theorem 4.3 and 4.5) which shows that 2-RF yields straight flow from Gaussian to a fairly large class of distributions including a general mixture of Gaussians. In the first rebuttal, we also provided a simple example where 2-RF is not straight, but as it turns out, in that case, the Monge map does not exist. Under the light of Theorem 4.5 (and Theorem 4.3), we believe it is hard to develop an example where a Monge map exists but the flow is not straight. In the discussion of the updated manuscript, we pose an open question "Does the existence of Monge map imply straightness of 2-RF?". We discussed this in our previous comments and Section 6 of the paper. Hopefully, the discussion will help clarify some of the concepts. Given the discussion period is coming to an end, we were wondering if you have any further questions that we could address so that we can improve the paper.

---

> ### Comment · Reviewer_ptxD · 2024-12-01
>
> P1. The new general result addresses my concern.
>
> P2. I do not mean that for any target distribution we have decoupling. I am just saying that a way to argue in Theorem 4.4, given that the problem does decouple for the given GM target distribution, would be to show that in each dimension we are straight. Can this idea simplify the proof of Theorem 4.4? (When I say that "1-Rectified flows are always straight in 1 dimension" I am assuming that the 1-Rectified flow exists. The counterexample you provide can not be rectified so it does not apply.)
>
> P3-P6: My concerns are addressed.
>
> I raise my score to 8 since this new general result appears to me as very important for the field of rectified flows.

---

> > ### Comment · Reviewer_ptxD · 2024-12-01
> >
> > Also, I believe the equation for $H_t$ in between lines 364 and 365 is missing a $t$ in front of $Z_1(z_0)$

---

> ### Author Response · Authors · 2024-12-01
>
> Thank you very much for raising the score and your valuable comments. We are glad that you found our results important.  Regarding your concern related to (P2), you are right that coordinate wise straightness would imply overall straightness. Therefore, showing coordinate wise straightness is enough to establish the straightness under the decoupling phenomenon. However, for general $\mu_1, \mu_2 \in \mathbb{R}^d$, the problem do not necessarily decouple. We have to change the coordinate system via a judicial rotation to allow us to decouple the coordinates. After that step, we essentially show that the problem is coordinatewise straight. The application of rotation is crucial in this proof.
>
> Also, thank you for pointing out the typo in lines 364-365. We will fix this in the updated version.

---

### Official Review · Reviewer_Apm1 · 2024-11-04

**Soundness:** 2
**Presentation:** 3
**Contribution:** 2
**Rating:** 6
**Confidence:** 4

**Summary:**

This submission presents an error analysis of rectified flow (RF) in terms of the Wasserstein distance. (1) It establishes an error bound for the Wasserstein distance between the target distribution and the sampling distribution generated by RF, based on the Lipschitz continuity of the vector fields and the average curvature of the flow curve, which measures its straightness. (2) For Gaussian mixtures, the authors demonstrate that 2-rectifications are sufficient to achieve a straightened flow.

**Strengths:**

The theoretical analysis is clear and intuitive. The choice of Wasserstein distance naturally aligns with the formulation of least squares (least action principle). The authors use that to control the Wasserstein distance between the target and sampling distributions based on the squared distance between the velocity fields. Additionally, this squared distance is further bounded by the average curvature through a Poincaré-type inequality. The authors also provide a rigorous justification for achieving strict straightness in 2-rectifications for Gaussian mixtures, a phenomenon that has also been observed numerically in more complex distributions.

**Weaknesses:**

The benefit of straightness in the rectified flow is not demonstrated. In the conclusion, the authors claim that "straight flows are desirable because they require fewer Euler discretization steps." However, there is no quantification of this gain or a clear link to straightness provided in the analysis.

Although the straightness of two-step rectified flow is proved for Gaussian mixtures, it remains unclear how this result can be extended to more general distributions.

**Questions:**

1. Could the authors provide a more detailed explanation or quantification of the computational advantages of achieving straightness in rectified flows? For instance, how does straightness impact the number of discretization steps required in practice?

2. How robust is the theoretical result for two-step rectified flow straightness when moving beyond Gaussian mixtures? Are there any indicators or preliminary results that suggest potential extensions to more general distributions?

---

> ### Author Response · Authors · 2024-11-18
>
> Thank you for your questions and constructive comments. We address your concerns point by point below.
>
> **(P1) The benefit of straightness in the rectified flow is not demonstrated. In the conclusion, the authors claim that "straight flows are desirable because they require fewer Euler discretization steps." However, there is no quantification of this gain or a clear link to straightness provided in the analysis.** -- We would like to point out that Theorem 3.5 (and Corollary 3.6) shows the dependency of $W_2$ distance on the straightness parameter $\gamma_{2,T}$. The discussion after Theorem 3.5 (line 304 - 315) clearly points out that $T = \Omega(\sqrt{\gamma_{2,T}/\epsilon})$ is enough to achieve desired error bound on $W_2$ distance. This shows that more straight flow requires less number of discretizations  (i.e., small $T$) steps resulting in faster generation time.
>
> **(P2) Although the straightness of two-step rectified flow is proved for Gaussian mixtures, it remains unclear how this result can be extended to more general distributions.**-- The benefit of rectified flow in terms of generation quality has been numerically demonstrated in past works like [1, 2]. In this, paper we provide a theoretical underpinning to this fact by connecting $W_2$ distance with the straightness parameter $\gamma_{2,T}$. [1, 2] also theoretically showed that the straightness parameter $S(\mathcal{Z})$ decreases (in some sense) with successive rectification. Therefore, we expect that $\gamma_{2,T}$ decreases for higher rectification. In Section 4, we actually theoretically show that $\gamma_{2,T} = 0$ (i.e., straight flow) after 2-rectification for some curated examples.
> In fact, while revising, we proved a more general result, i.e.,  under very mild assumptions on the target distribution, the parameter $\gamma_{2,T} = 0$ after 2-rectification (Theorem 4.5 and Theorem 4.3). An informal version of the Theorem 4.5 is shown below:
>
> *_Theorem 4.5 (informal version)_: Assume $X_0 \sim N(0, I_d)$ and the target data $X_1 \sim \rho_1$ (independent of $X_0$) with finite 1st moment, i.e., $\mathbb{E} \Vert X_1\Vert_2< \infty$. Also, assume that the ODE (3) is non-explosive (essentially solutions $Z_t$ do not diverge to infinity). Then, the resulting rectified coupling $(Z_0,Z_1):= \texttt{Rect}(X_0, X_1)$ under the aforementioned conditions is  a straight coupling*
>
> This shows that we do not need any discretization after 2-rectification.
>
>  **(P3) Could the authors provide a more detailed explanation or quantification of the computational advantages of achieving straightness in rectified flows? For instance, how does straightness impact the number of discretization steps required in practice?** --  Numerical experiments in [1,2] have already demonstrated the benefit of rectified flow and we also discuss those in the 2nd paragraph of Page 1 (line 44-51). Essentially, rectification aims to straighten a potentially curved flow between two distributions, thus reducing the transport cost. This allows the rectified flow to generate higher-quality data within fewer steps. We also discuss about the dependency of discretization steps $T$ with the straightness parameter $\gamma_{2,T}$ in the discussion after Theorem 3.5 (Line 304-316).
>
>  **(P4) How robust is the theoretical result for two-step rectified flow straightness when moving beyond Gaussian mixtures? Are there any indicators or preliminary results that suggest potential extensions to more general distributions?**-- Thank you for your question. Yes, the result actually holds in more generality and we proved this while revising the paper. Please refer to Theorem 4.5 (and Theorem 4.3) of the updated version.
>
> Note that, target distribution does not need to have a pdf, it only needs to have a finite 1st moment. Also, the non-explosive condition is fairly mild and is enjoyed by a large class of distributions including a general mixture of Gaussians.
>
> [1] X. Liu, X. Zhang, J. Ma, J. Peng, et al., “Instaflow: One step is enough for high-quality diffusion-based
> text-to-image generation,” in The Twelfth International Conference on Learning Representations, 2024.
>
> [2] X. Liu, C. Gong, and Q. Liu, “Flow straight and fast: Learning to generate and transfer data with
> rectified flow,” in The Eleventh International Conference on Learning Representations, 2023.

---

> > ### Comment · Reviewer_Apm1 · 2024-11-27
> >
> > I appreciate the authors' efforts in addressing my questions, particularly in extending the existing results to more general distributions. I have raised my score accordingly. However, the authors could provide further discussion on the non-explosiveness of the ODE under the assumptions of the new theorem.

---

> > > ### Author Response · Authors · 2024-11-27
> > >
> > > Thank you very much for your comment and for raising the score.  We will add more discussion on the non-explosivity of the ODE either after the assumption or we will refer to the appropriate discussion section in appendix.

---

### Author Response · Authors · 2024-11-18
**Revised version uploaded and A new general straightness result is added**

We thank all the reviewers for their suggestions and constructive comments. We have revised the paper and uploaded the new version. More importantly, In Section 4, we have added two new general results (Theorem 4.3 and Theorem 4.5) that shows that **1-RF yields a straight coupling for a large class of distributions (which extends beyond Gaussian mixtures).** We would like to emphasize the generality of this result, which, to the best of our knowledge, has not been previously reported in the literature. An informal version of Theorem 4.5 is stated below.

*_Theorem 4.5 (informal version)_: Assume $X_0 \sim N(0, I_d)$ and the target data $X_1 \sim \rho_1$ (independent of $X_0$) with finite 1st moment, i.e., $\mathbb{E} \Vert X_1\Vert_2< \infty$. Also, assume that the ODE (3) is non-explosive (essentially solutions $Z_t$ do not diverge to infinity). Then, the resulting rectified coupling $(Z_0,Z_1):= \texttt{Rect}(X_0, X_1)$ under the aforementioned conditions is  a straight coupling*

Note that, in the theorem, target distribution does not need to have a pdf, it only needs to have a finite 1st moment. Also, the non-explosive condition is fairly mild and is enjoyed by a large class of distributions including a general mixture of Gaussians. Therefore, the current results provide a general straightness result for a large class of rectified flow. We have also added recommended simulation experiments that corroborate our theoretical claims.

---

### Author Response · Authors · 2024-11-25
**Revised(minor) version uploaded: Slight change in paper title and minor typo corrections**

We want to sincerely thank all the reviewers again for handling our submission. We wanted to point out that we have made a slight modification of the paper title to underscore the new straightness results (Theorem 4.3 and Theorem 4.5) mentioned in the rebuttal. The current title reads "2-RECTIFICATIONS ARE ENOUGH FOR STRAIGHT FLOWS: A THEORETICAL INSIGHT INTO WASSERSTEIN CONVERGENCE." We could not change the title in the Openreview directly as we could not find the option. However, we will change it during the camera-ready submission period if accepted. In addition, we have also addressed a few typographical errors and minor issues that were pointed out by the reviewers.

Regards,

---

### Note · Authors · 2025-02-28

**Comment:**

We have found a bug in the proof of one of our main results. We have resolved this issue by formulating necessary and sufficient conditions for our result to hold. Unfortunately, these conditions are fairly abstract, and we can only verify them empirically.  More work is necessary. Therefore, we do not believe our results are ready for publication yet.

**Withdrawal Confirmation:**

I have read and agree with the venue's withdrawal policy on behalf of myself and my co-authors.

---

### Meta-Review · Area_Chair_8RFH · 2024-12-18

**Metareview:**

This paper is on the topic of rectified flow. It makes two theoretical contributions to the study of rectified flow. First, it provides a theoretical analysis to quantify the Wasserstein error of a trained rectified flow model. Second, it shows, under proper assumptions, 2 rectifications are sufficient to achieve a straight flow. All reviewers agree the paper is well written and present the theoretical contributions clearly. Two reviewers point out that the experiments are insufficient to illustrate the theoretical results. One reviewer questions the tightness of the theoretical bound. The authors provide satisfactory answers to these questions. Overall, this is a nice contribution to the rectified flow framework.

**Additional Comments On Reviewer Discussion:**

The reviewers raise several questions on the technical results of the paper. The authors clarify these points and make modifications to address the comments. The reviewers are convinced and raise points accordingly.

---

### Decision · Program_Chairs · 2025-01-22

Accept (Poster)